# FairGRPO: Fair Reinforcement Learning for Equitable Clinical Reasoning

## Abstract

Medical artificial intelligence systems have achieved remarkable diagnostic capabilities, yet they consistently exhibit performance disparities across demographic groups, causing real-world harm to underrepresented populations. While recent multimodal reasoning foundation models have advanced clinical diagnosis through integrated analysis of diverse medical data, reasoning trainings via reinforcement learning inherit and often amplify biases present in training datasets dominated by majority populations. We introduce **Fairness-aware Group Relative Policy Optimization (FairGRPO)**, a hierarchical reinforcement learning approach that promotes equitable learning across heterogeneous clinical populations. FairGRPO employs adaptive importance weighting of advantages based on representation, task difficulty, and data source. To address the common issue of missing demographic labels in the clinical domain, we further employ unsupervised clustering, which automatically discovers latent demographic groups when labels are unavailable. Through comprehensive experiments across 7 clinical diagnostic datasets spanning 5 clinical modalities across X-ray, CT scan, dermoscropy, mammography and ultrasound, we demonstrate that FairGRPO reduces predictive parity by 27.2% against all vanilla and bias mitigated RL baselines, while improving F1 score by 12.49%. Furthermore, training dynamics analysis reveals that FairGRPO progressively improves fairness throughout optimization, while baseline RL methods exhibit deteriorating fairness as training progresses. Based on FairGRPO, we release **FairMedGemma-4B**, a fairness-aware clinical VLLM that achieves state-of-the-art performance while demonstrating significantly reduced disparities across demographic groups. Our code, models, and fairness evaluation framework are publicly available at this anonymous link.

## 1 Introduction

Medical artificial intelligence (AI) has demonstrated strong capabilities in processing vast amounts of clinical data with both accuracy and efficiency (Rajpurkar et al., 2022; Shuja et al., 2024). These systems have shown particular promise in detecting subtle health indicators that may escape human observation, substantially enhancing diagnostic precision while reducing healthcare costs (Dai et al., 2025a; Sun et al., 2022). Recent advances in vision large language models (VLLMs) have further expanded these capabilities, enabling integrated analysis across diverse clinical modalities including imaging, time series, and textual records (Cui et al., 2024; Dai et al., 2025b; Zhang et al., 2024; Zhu et al., 2024).

However, beneath these impressive achievements lies a fundamental challenge that undermines the equitable deployment of AI in healthcare. Medical AI systems can consistently exhibit troubling performance disparities across demographic subpopulations. Studies have revealed that clinical datasets are overwhelmingly skewed toward majority groups, whether defined by race, gender, age, or socioeconomic status (Larrazabal et al., 2020; Obermeyer et al., 2019; Liang et al., 2021; Thakur et al., 2023). State-of-the-art (SOTA) classifiers demonstrate significant true positive rate (TPR) disparities across all clinical tasks, datasets, and demographic subgroups (Seyyed-Kalantari et al., 2021a;b). Such systematic biases not only perpetuate healthcare inequalities but also erode trust in AI-assisted diagnosis, particularly among underserved communities who stand to benefit most from improved healthcare access (Sagona et al., 2025).

During training, conventional optimization approaches naturally favor well-represented populations, as they contribute more gradient updates and dominate the loss landscape (Stiglic et al., 2020; Kumarakulasinghe et al., 2020). This creates a pernicious feedback loop: models become increasingly specialized for majority populations while performance on minority groups stagnates or even degrades. Furthermore, the heterogeneous nature of clinical data spanning multiple specialties, modalities, and patient demographics, can exacerbate these disparities as different groups may require fundamentally different diagnostic considerations (Ghanvatkar & Rajan, 2023; Cui et al., 2023).

Current approaches to mitigating bias in medical AI typically rely on data augmentation, reweighting schemes, or post-hoc calibration (Teng et al., 2022; Khan et al., 2023; Mehta et al., 2024). However, the emergence of reasoning-capable vision LLMs introduces unique challenges that existing methods cannot adequately address. For instance, fairness-aware optimization techniques like group distributionally robust optimization (DRO) (Sagawa et al., 2019) were designed for discriminative models with *fixed* output spaces and cannot be directly applied to the *generative, multi-step reasoning processes* characteristic of modern LLMs. Furthermore, while reinforcement learning (RL) has revolutionized LLM alignment for helpfulness and harmlessness (Ouyang et al., 2022; Bai et al., 2022), its application to fairness in medical reasoning remains unexplored. Fairness in medical settings can be particularly challenging given how disease diagnosis typically relies on the comprehensive analysis of and reasoning between multiple symptoms, mismatch in data availability across different domains (e.g. abundance in X-ray but lacking in ultrasound) and how data collection is skewed towards those with access to healthcare. The complex interplay between reward modeling, advantage estimation, and demographic disparities in the context of clinical reasoning presents a novel optimization challenge that requires fundamentally new approaches.

To close this gap, we introduce **Fairness-aware Group Relative Policy Optimization (Fair-GRPO)**: a hierarchical RL approach that promotes equitable learning across heterogeneous clinical populations. Our work makes two primary contributions:

1. We propose one of the first fair RL algorithm, **FairGRPO**, that employs *adaptive importance weighting* based on demographic representation and task difficulty, ensuring that minority groups equitable learning signals. Our empirical evaluation demonstrates that FairGRPO consistently improves both overall performance and fairness metrics. Specifically, FairGRPO reduces predictive parity by 27.2% against all vanilla and bias mitigated RL baselines, while improving F1 score by 12.49%. Furthermore, training dynamics analysis reveals that FairGRPO improves fairness of the model during the training process, while other RL algorithms exhibit a deterioration of fairness as the training progresses.

2. Based on FairGRPO, we train and release **FairMedGemma-4B**, a fairness-aware vision clinical model based on MedGemma that excel across 7 clinical datasets spanning 5 clinical modalities. FairMedGemma not only achieves SOTA performance on standard benchmarks but also demonstrates significantly reduced disparities across demographic groups, advancing the development of equitable AI-assisted diagnosis. To the best of our knowledge, FairMedGemma represents the first publicly available clinical VLLM explicitly optimized for demographic fairness through reinforcement learning.

Finally, we publicly release our models, training pipeline, and comprehensive fairness evaluation metrics to facilitate reproducible research in equitable medical AI. By addressing fairness as a fundamental optimization objective rather than a post-hoc consideration, our work establishes a new paradigm for developing clinical AI systems that serve all populations equitably.

## 2 RELATED WORK

**Fairness in Unimodal and Multimodal Clinical Diagnosis.** While unimodal clinical diagnosis leverages single data sources (e.g., images (Khan et al., 2023; Mehta et al., 2024) or tabular data (Dehghani et al., 2024; Röösli et al., 2022)), multimodal methods fuse multiple modalities to learn richer representations, consistently outperforming unimodal approaches (Liang et al., 2024; Dai et al., 2025c; AlSaad et al., 2024) across radiology (Yildirim et al., 2024), psychiatry (Lee et al., 2024; Cheong et al., 2025a), and ophthalmology (Luo et al., 2024). The increasing adoption of foundation models in healthcare (Dai et al., 2025c; Jin et al., 2024; Luo et al., 2024) amplifies fairness challenges, as integrating multiple knowledge sources can exacerbate biases across fused modalities. Fairness in ML, broadly categorized into group or individual fairness (Mehrabi et al.,

2021; Hort et al., 2024; Waller et al., 2025), has been primarily studied in unimodal settings such as chest radiographs (Khan et al., 2023; Mehta et al., 2024), EEG data (Kurbatskaya et al., 2023; Kwok et al., 2025), or EHR data (Dehghani et al., 2024; Röösli et al., 2022). Recent work has begun investigating multimodal fairness in healthcare (Cheong et al., 2024; Luo et al., 2024; Wang et al., 2024; Cheong et al., 2025b), but existing studies typically focus on single clinical tasks, such as depression detection (Cheong et al., 2024), kidney tumor segmentation (Afzal et al., 2023), or glaucoma detection (Luo et al., 2024). Our work presents the first attempt to evaluate fairness on a model trained across multiple clinical tasks and domains simultaneously.

**Fairness in Reinforcement Learning.** Reinforcement learning (RL) methods which typically attempt to maximize the reward of an agent as defined by a specific objective may neglect fairness considerations (Jabbari et al., 2017; Smith et al., 2023). Recent advances in critic-free RL algorithms for LLMs, such as GRPO (Shao et al., 2024), RLOO (Ahmadian et al., 2024), and REINFORCE++ (Hu, 2025), have demonstrated remarkable success in aligning language models without requiring value function estimation. However, these methods lack mechanisms to address fairness across heterogeneous populations. Traditional fairness in RL can be categorized into single- or multi-agent settings (Reuel & Ma, 2024; Yang et al., 2023; Sahoo et al., 2024), with resampling Puyol-Antón et al. (2021) and Group DRO Sagawa et al. (2019) being two popular fairness mitigation methods. To the best of our knowledge, however, none of the current works address the fairness challenge in critic-free RL optimization of VLLMs, where the computational requirements and multi-step reasoning processes present unique challenges distinct from traditional RL settings. Our work bridges this gap by extending GRPO with fairness-aware mechanisms specifically designed for the requirements of medical VLLMs.

**Fairness in ML and Large Language Models.** Recent multimodal LLMs such as Qwen-2.5-VL (Bai et al., 2025) and domain-specific models like MedGemma (Sellergren et al., 2025) have demonstrated impressive clinical reasoning capabilities, yet their fairness properties remain largely unexplored. While models like DeepSeek-R1 (Guo et al., 2025) have advanced reasoning through reinforcement learning, they lack mechanisms to ensure equitable performance across demographic groups. Existing fairness works in healthcare FMs (Khan et al., 2023; Jin et al., 2024; Luo et al., 2024) have focused on predictive bias in unimodal models. Khan et al. (2023) revealed consistent under-performance for female patients, while Luo et al. (2024) proposed optimal-transport approaches for performance-fairness tradeoffs. However, these methods cannot address the unique challenges of reasoning-capable VLLMs, where multi-step reasoning and reinforcement learning create new pathways for bias amplification. Our work is the first to tackle fairness in critic-free RL training for multimodal clinical reasoning models.

## 3 METHOD

Medical AI systems often exhibit performance disparities across demographic subpopulations, reflecting biases inherent in training data distributions (Luo et al., 2024; Khan et al., 2023). While Group Relative Policy Optimization (GRPO) has demonstrated success in language model alignment through within-group reward normalization, it lacks mechanisms to address systematic subgroup imbalances across heterogeneous populations. We introduce FairGRPO, a hierarchical scaling approach that promotes equitable learning by adaptively weighting contributions from different domains and demographic groups based on their demographic information and difficulty measured via model performance.

**Background: Group Relative Policy Optimization (GRPO).** GRPO operates by normalizing rewards within groups of responses to identical prompts, eliminating the need for value function estimation. For a prompt $q$ generating response group $G_{(q,t)}$ at iteration $t$, each response $o_{(q,i,t)}$ receives reward $r_{(q,i,t)}$. The advantage is computed as $\hat{A}^{\text{GRPO}}_{(q,i,t)} = \frac{r_{(q,i,t)} - \hat{\mu}_{G_{(q,t)}}}{\hat{\sigma}_{G_{(q,t)}} + \varepsilon}$, ensuring zero mean and unit variance within each response group. This normalization enables fair comparison among responses to the same prompt but treats all prompts equally, regardless of their source domain or demographic representation.

**The Fairness Challenge.** Consider a training dataset where prompts originate from different domains $g \in \mathcal{G}$ and are associated with demographic groups $d \in \mathcal{D}_{\text{demo}}$. Each prompt $q$ at iteration $t$ belongs to exactly one domain $g_{(q,t)}$ and one demographic group $d_{(q,t)}$.

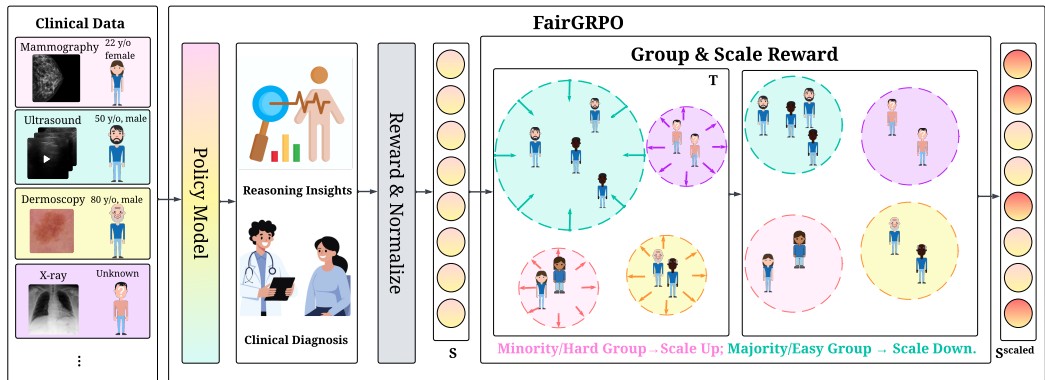

Figure 1: **FairGRPO Training Pipeline.** Our method addresses fairness disparities by adaptively scaling rewards based on demographic representation and task difficulty. Starting with medical data containing both labeled demographic information and unlabeled samples, the policy model generates multiple responses for each prompt, producing both reasoning insights and clinical diagnoses. These responses are evaluated and assigned rewards. FairGRPO then groups the rewards by explicit demographic groups where available. For samples with unavailable demographic information, we employ K-means clustering to discover implicit groups. Then, minority or challenging groups receive amplified learning signals through inverse temperature scaling, while majority or well-represented groups are scaled down. This ensures that the model learns equitably from all subpopulations, preventing the typical bias toward majority groups that occurs in standard training.

Standard GRPO optimization naturally favors well-represented domain-demographic pairs, as they contribute more gradient updates. This creates a feedback loop where the model becomes increasingly specialized for majority populations while performance on minority groups stagnates. FairGRPO breaks this cycle through adaptive importance weighting that inversely correlates with group representation and performance.

**Hierarchical Scaling Framework.** FairGRPO implements a three-stage process that transforms GRPO's uniform treatment into demographically-aware optimization:

*(i) Normalization:* We first apply standard GRPO normalization to obtain $s_{(q,i,t)} = \frac{r_{(q,i,t)} - \hat{\mu}_{G_{(q,t)}}}{\hat{\sigma}_{G_{(q,t)}} + \varepsilon}$.

*(ii) Group Discovery:* In medical datasets, demographic labels may be incomplete or unavailable for certain samples. We define *explicit groups* as those with *labeled* demographic attributes such as age or gender while *implicit groups* are latent subpopulations discovered through unsupervised clustering when such labels are missing. To identify implicit groups, we leverage the model's performance patterns: within each domain $g$, we construct feature vectors $\mathbf{v}_q \in \mathbb{R}^{|G_{(q,t)}|}$ for each unlabeled prompt $q$, where each dimension represents the raw reward from a different rollout. In GRPO, a rollout refers to a single generated response for a given prompt, with multiple rollouts per prompt enabling reward normalization across response variations. For instance, a chest X-ray prompt without demographic labels might generate 5 rollouts with rewards [0.2, 0.8, 0.7, 0.9, 0.3], forming its feature vector.

This reward-based representation offers two key advantages over traditional feature extraction methods. First, it provides exceptional computational efficiency, requiring only a vector of length equal to the number of rollouts rather than high-dimensional CNN or ViT embeddings. Second, and more importantly, it directly captures task-specific difficulty patterns rather than input-level similarities. While visual features might group images by appearance, our approach groups samples by their inherent diagnostic challenge to the model, ensuring that cases with similar learning difficulties receive similar treatment regardless of their visual characteristics. K-means clustering then groups prompts with similar reward distributions, where common, well-represented cases typically form larger clusters with consistently higher rewards, while rare or challenging cases naturally form smaller clusters with lower or more variable rewards. The optimal number of clusters is determined automatically via the elbow method (Thorndike, 1953) in alignment with existing works (Weng et al., 2021; Cai et al., 2025). Crucially, because our scaling mechanism inversely weights the reward by cluster size

Table 1: **List of Experimental Datasets.** We use 7 datasets across 5 clinical modalities. The performance metrics are an unweighted average of datasets across classes, as described in Sec. 4.1.

| Dataset | # samples | Clinical domain | Modality | Labels | Demographics |
|---|---|---|---|---|---|
| CheXpert | 212K | Radiology | Chest X-ray | Atelectasis, Cardiomegaly, Consolidation, Edema, Enlarged Cardiomediastinum, Fracture, Lung Lesion, Lung Opacity, Pleural Effusion, Pneumonia, Pneumothorax, Pleural Other, Support Devices, No Finding | Age, Sex |
| Hemorrhage | 2.5K | Radiology | CT | No Hemorrhage, Has Hemorrhage | Age, Sex |
| VinDr-Mammo | 20K | Radiology, Oncology | Mammography | BI-RAD 1-5 | Age |
| ISIC-2020 | 33K | Dermatology, Oncology | Dermoscopy | Malignant, Benign | Age, Sex |
| HAM10000 | 10K | Dermatology, Oncology | Dermoscopy | Melanoma (MEL), Nevus (NV), Basal Cell Carcinoma (BCC), Actinic Keratosis/Intraepithelial Carcinoma (AKIEC), Other (OTHER) | Age, Sex |
| PAD-UFES-20 | 2.3K | Dermatology, Oncology | Dermoscopy | Melanoma (MEL), Nevus (NV), Basal Cell Carcinoma (BCC), Actinic Keratosis/Intraepithelial Carcinoma (AKIEC), Other (OTHER) | Age, Sex |
| COVID-BLUES | 362 | Radiology | Ultrasound | Has COVID, No COVID | Age |

and performance as shown in Equations 1, these smaller clusters representing rarer or more difficult cases receive amplified learning signals, ensuring that even unlabeled minority subpopulations benefit from our fairness-aware optimization.

*(iii) Demographic Group Based Reward Scaling:* We compute hierarchical temperature factors that capture both representation and difficulty. At the domain and group level, this is represented by:

$$T_{(g,t)} = \sqrt{N_{(g,t)}} \cdot \bar{r}_{(g,t)}, T_{(\gamma,g,t)} = \sqrt{N_{(\gamma,g,t)}} \cdot \bar{r}_{(\gamma,g,t)}. \tag{1}$$

respectively for group $\gamma$ (explicit or implicit) in domain $g$. $N_{(g,t)}$ counts samples in domain $g$ and $\bar{r}_{(g,t)}$ represents the domain's mean raw reward. The normalized rewards undergo inverse temperature scaling:

$$s_{(q,i,t)}^{\text{scaled}} = \frac{s_{(q,i,t)}}{\max(T_{(g_{(q,t)},t)} \cdot T_{(\gamma_{(q,t)},g_{(q,t)},t)}, \varepsilon)}, \tag{2}$$

thus amplifying signals from underrepresented or challenging groups while attenuating those from dominant populations. Lastly, following (Schulman et al., 2017), we renormalize the advantage to zero mean and unit variance with $\hat{A}_{(q,i,t)}^{\text{FairGRPO}} = \frac{s_{(q,i,t)}^{\text{scaled}}}{\sigma_{\text{batch}}}$, where $\sigma_{\text{batch}}$ denotes the standard deviation across all scaled rewards in the current batch.

**Training Objective.** FairGRPO retains GRPO's policy gradient formulation with clipped importance sampling:

$$J_{\text{FairGRPO}}(\theta) = \mathbb{E}_{q,o} \left[ \sum_{k=1}^{n_o} \min \left( \varphi_k(\theta) \hat{A}^{\text{FairGRPO}}, \text{clip}(\varphi_k(\theta), 1 \pm \varepsilon) \hat{A}^{\text{FairGRPO}} \right) - \beta D_{\text{KL}}(\pi_\theta \| \pi_{\text{ref}}) \right],$$

where $\varphi_k(\theta)$ represents the importance ratio at token $k$, and the advantage now incorporates fairness-aware scaling.

**Reward Design.** FairGRPO works with arbitrary reward designs. In the experiment of this work, we employ a standard accuracy reward where the model gets a reward of 1 if the final answer is correct, and a reward of 0 if the answer is incorrect.

## 4 EXPERIMENTS

### 4.1 DATASETS & EXPERIMENTAL SETUP

We design experiments to comprehensively evaluate FairGRPO's ability to improve both performance and fairness across diverse clinical subpopulations. Our experimental framework addresses the following three key research questions:

**RQ1: How does FairGRPO perform compared to other RL methods?** Given the distinct training procedures across multimodal reasoning LLM methods, we benchmark FairGRPO against RL baselines including GRPO (Shao et al., 2024), RLOO Ahmadian et al. (2024) and REINFORCE++ (Hu, 2025). These methods represent the current state-of-the-art in critic-free reinforcement learning for LLMs. To compare our methods against other fairness mitigation algorithms, we re-implement

Table 2: **RQ1: Fairness and performance metrics comparison against RL and fairness mitigation baselines.** For fairness metrics, lower values are better and are indicated by ↓. For performance and combined metrics, higher values are better and are indicated by ↑. Bold values indicate the best result in each column for each model separately. **FairGRPO**$_{ND}$ is the ablation of **FairGRPO** where the model does not have access to the ground truth demographic information, and the groups are inferred entirely via clustering. We release **MedGemma** trained with **FairGRPO** as **FairMedGemma**. Detailed per dataset metrics are included in App. Tab. 5-17.

| Training Method | Fairness Metrics | | | | | | | Perf. Metrics | | Combined | |
|---|---|---|---|---|---|---|---|---|---|---|---|
| | PP ↓ | EOD ↓ | FPR$_{Diff}$ ↓ | $\sigma_{F1}$ ↓ | $\Delta$F1 ↓ | $\sigma_{Acc}$ ↓ | $\Delta$Acc ↓ | Acc ↑ | F1 ↑ | Acc$_{ES}$ ↑ | F1$_{ES}$ ↑ |
| *Qwen-2.5-VL-7B* | | | | | | | | | | | |
| Re++ (Hu, 2025) | 15.18 | 7.788 | 6.233 | .0322 | .0650 | 4.706 | 9.613 | 75.32 | .2612 | 71.93 | .2531 |
| RLOO (Ahmadian et al., 2024) | 21.73 | 6.577 | 5.115 | .0326 | .0705 | 5.098 | 10.56 | 79.67 | .2479 | 75.80 | .2400 |
| GRPO (Shao et al., 2024) | **11.39** | 9.091 | 4.607 | .0463 | .0973 | 4.676 | 9.433 | 80.45 | .2550 | 76.85 | .2437 |
| GRPO+RS (Puyol-Antón et al., 2021) | 21.56 | 8.091 | 4.961 | .0316 | .0636 | **3.967** | **8.113** | 73.99 | **.2657** | 70.57 | .2576 |
| GRPO+DRO (Sagawa et al., 2019) | 14.51 | 7.413 | 7.417 | .0326 | .0654 | 5.621 | 11.50 | 75.10 | .2586 | 71.10 | .2504 |
| **FairGRPO** | 16.80 | **5.546** | **4.391** | **.0229** | **.0452** | 4.410 | 8.934 | **80.75** | .2647 | **77.34** | **.2588** |
| *MedGemma-4B* | | | | | | | | | | | |
| Re++ (Hu, 2025) | 20.99 | 8.749 | 5.616 | .0518 | .1033 | 4.317 | 8.821 | 78.60 | .2978 | 75.35 | .2831 |
| RLOO (Ahmadian et al., 2024) | 23.68 | 10.37 | 5.513 | .0600 | .1170 | 4.336 | 8.837 | 80.62 | .3047 | 77.27 | .2875 |
| GRPO (Shao et al., 2024) | 22.42 | 6.476 | 4.820 | .0418 | .0795 | 4.171 | 8.546 | 80.02 | .3123 | 76.82 | .2998 |
| GRPO+RS (Puyol-Antón et al., 2021) | 23.76 | 6.664 | **3.481** | .0433 | .0835 | 4.051 | 8.386 | 80.76 | .2843 | 77.62 | .2725 |
| GRPO+DRO (Sagawa et al., 2019) | 16.04 | 7.367 | 4.985 | .0447 | .0871 | 4.362 | 8.960 | 81.19 | .3271 | 77.80 | .3009 |
| **FairGRPO**$_{ND}$ | 25.15 | 11.56 | 5.692 | .0547 | .1067 | **3.613** | **7.214** | 79.23 | **.3513** | 76.47 | **.3331** |
| **FairGRPO (FairMedGemma)** | **11.67** | **6.663** | 5.330 | **.0383** | **.0721** | 4.081 | 8.455 | **81.83** | .3218 | **78.62** | .3100 |

popular bias mitigation method, namely Group DRO (Sagawa et al., 2019) and Resampling Puyol-Antón et al. (2021), on top of GRPO. We employ a suite of fairness metrics, including Equal Opportunity Difference, Equalized Odds, and Predictive Parity, alongside standard performance metrics (F1, accuracy) as detailed in Appendix A.1, which ensures we capture both the utility and equity dimensions of model performance.

**RQ2: How do fairness metrics evolve during training?** Understanding the dynamics of fairness during optimization is crucial for guiding the future training strategies of VLLMs. We track the progression of fairness by measuring the maximum F1 score difference across the different demographic subgroups at 5-step intervals throughout training. In this experiment, we aim to monitor whether FairGRPO's hierarchical scaling mechanism consistently reduces disparities or merely achieves fairness at convergence. By comparing these trajectories against standard GRPO, we can assess whether our adaptive weighting strategy changes the optimization landscape.

**RQ3: How does performance vary across individual demographic groups?** Beyond aggregated fairness metrics, we analyze group-specific outcomes by examining average F1 scores for each demographic subpopulation. This analysis reveals whether improvements are uniformly distributed or concentrated in specific subgroups, and crucially, whether minority group gains come at the expense of majority group performance.

To demonstrate generalizability across architectures and ensure robust evaluation, we implement FairGRPO on two widely used VLLMs: Qwen-2.5-VL-7B (Bai et al., 2025) and MedGemma-4B (Sellergren et al., 2025). Following the standard multitask instruction tuning paradigm in both works, we initialize from pretrained weights and perform unified finetuning across all 7 clinical datasets simultaneously in a single training run, mirroring real-world deployment where models must handle diverse clinical tasks without dataset-specific adaptation. All experiments utilize 4 NVIDIA H200 GPUs. Hyperparameters and training configurations are detailed in Appendix A.

**Datasets.** To ensure our methods work across different clinical datasets, we evaluate the models via 7 public datasets, including CheXpert (Irvin et al., 2019), COVID-BLUES (Wiedemann et al., 2021), VinDr-Mammo (Nguyen et al., 2021), ISIC-2020 (Rotemberg et al., 2021), HAM10000 (Tschandl et al., 2018b), PAD-UFES-20 (Pacheco et al., 2020) and Hemorrhage (Hssayeni et al., 2020), with a total of 280.2K samples, as summarized in Tab. 1 and detailed in Appendix B.

**Demographic Groups.** We define demographic groups consistently across all datasets to ensure fair comparison. For gender, we use the patient gender as recorded in each dataset. For age, we create four groups using 25-year bins: a1 for ages 18-25, a2 for ages 26-50, a3 for ages 51-75, and a4 for ages 76 and above. This standardized binning strategy allows us to analyze fairness patterns

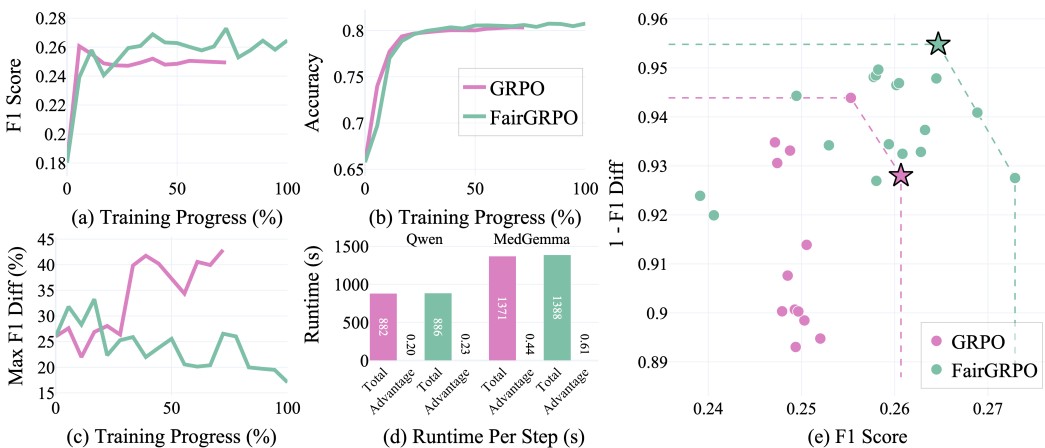

Figure 2: Training dynamics comparison between GRPO and FairGRPO on clinical classification tasks. **(a) F1 Score:** FairGRPO achieves higher F1 scores throughout training, reaching 0.265 compared to GRPO's plateau at 0.250. **(b) Accuracy:** Both methods converge to similar accuracy levels, with FairGRPO demonstrating slightly higher final accuracy. **(c) F1 Diff:** FairGRPO substantially reduces demographic performance disparities, achieving around 57% reduction in F1 difference by explicitly optimizing for fairness during training. **(e) Per Step Runtime of the Models:** We run the model using the setup described in Sec. 4.1. The reward calculation for all methods are less than 0.1% of the total runtime, showing it adds negligible overhead to the training process. **(e) Performance-Fairness Tradeoff:** We compare the validation F1 score and reversed F1 difference (1-F1 Diff) of different steps throughout a single training run. Pareto frontier is plotted to illustrate the points where the mdoel achieves the best tradeoff performance between F1 score and fairness. The starred point is the final model reported in Tab. 18. FairGRPO achieves superior Pareto optimality, simultaneously improving both performance and fairness compared to GRPO's best checkpoint.

across datasets with varying age distributions while maintaining sufficient sample sizes within each demographic group for meaningful statistical analysis.

**Evaluation Metrics.** For performance assessment, we use hierarchical averaging of F1 scores across classes, demographic groups, and datasets to prevent any single component from dominating the evaluation. For fairness evaluation, following (Hort et al., 2024), we measure popular fairness metrics including Equal Opportunity Difference (EOD), Predictive Parity (PP), and performance variance metrics ($\sigma_{\text{F1}}$, $\Delta$F1) to capture equity across demographic groups. To balance the fairness-utility tradeoff, following (Jin et al., 2024), we adopt Equity Scaling metrics (F1$_{\text{ES}}$, Acc$_{\text{ES}}$) that penalize models achieving high average performance at the cost of large demographic disparities. Full mathematical definitions and detailed descriptions of all metrics are provided in Appendix A.1.

### 4.2 RQ1: How does FairGRPO perform compared to other RL methods?

We trained multimodal LLMs with FairGRPO and compare it against baseline RL algorithms, and recorded results in Tab. 18. Overall, FairGRPO outperforms the baseline in both fairness metrics and performance metrics on both multimodal LLMs. In particular, FairGRPO outperforms classical bias mitigation methods in both fairness and diagnosis performance, thanks to its dynamic integration with the RL training method. On MedGemma, it reaches a 27.2% better predictive parity than the best fairness mitigation method Group DRO, reimplemented on top of GRPO. Compared to the best RL training method, EOD improves by 23.8% on MedGemma, and by 15.7% on Qwen-2.5-VL. Compared with all baselines, the maximum F1 gap decreases by 28.9% on Qwen-2.5-VL and by 8.37% on MedGemma. This shows FairGRPO's superiority in the field of improving fairness.

Furthermore, the FairGRPO$_{ND}$ performance demonstrates that FairGRPO improves fairness and performance even when no demographic information is passed during training, thanks to the latent group discovery algorithm via clustering. Compared with all baselines, FairGRPO$_{ND}$ achieves a 10.81% improvement in the Maximum Accuracy Gap, a 13.38% improvement in the standard

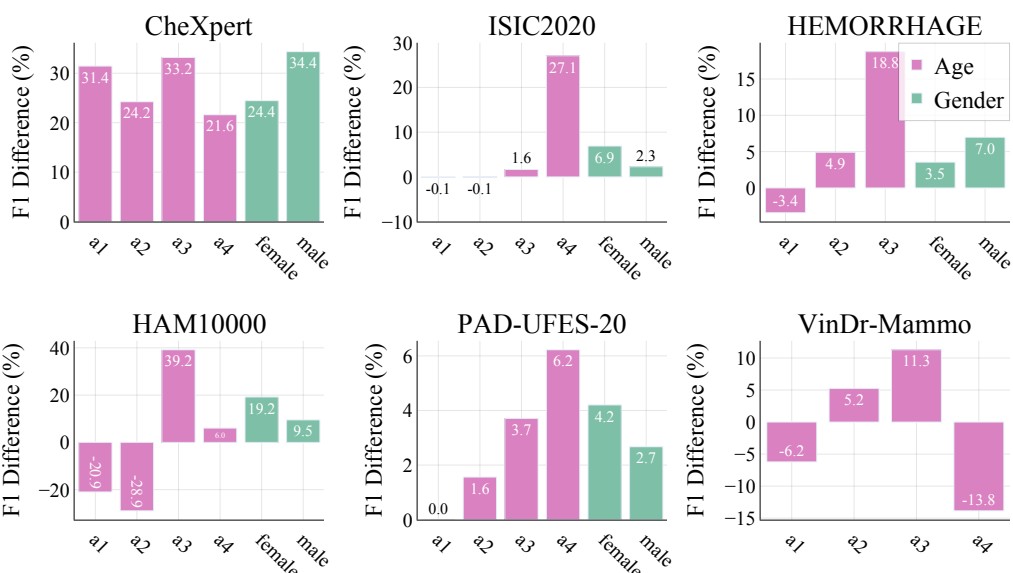

Figure 3: **F1 score differences between FairGRPO and GRPO across demographic groups on MedGemma.** Each bar represents the F1 score difference from the population mean for specific demographic subgroups, where a positive value means FairGRPO performs better for the given demographic group. The four age groups are binned as described in Sec. 4.1. In general, Fair-GRPO consistently demonstrates better performance for 25 out of the 33 demographic groups across datasets, which includes both majority and minority groups. Raw performance results are included in App. Tab. 4.

deviation of accuracy. $FairGRPO_{ND}$ shows particularly strong performance in F1, possibly due to the fact that its latent clustering aligns better with downstream tasks, as evidenced by its 12.49% improvement on F1, and 11.11% improvement in $F1_{ES}$ on MedGemma.

### 4.3 RQ2: How do fairness metrics evolve during training?

We recorded how the performance and fairness of FairGRPO and GRPO progress throughout a standard training run. As shown in Fig 2(c), although both methods improve the model's performance, the F1 difference for FairGRPO is lower than that of GRPO, and the gap between the two methods constantly increase as the runtime increases. In addition, Fig 2(a) and Fig2(b) show that the F1 score in FairGRPO is higher than that of GRPO, and the accuracy for both methods is almost the same. Fig 2(e) demonstrates that FairGRPO expands the empirical Pareto frontier relative to GRPO. Throughout the training process, the model provides multiple optimal checkpoints at various fairness-performance tradeoffs, all at better and more balanced Pareto points than GRPO.

**Runtime Efficiency.** Fig 2(d) shows that FairGRPO and GRPO's runtime per step is close on both Qwen2.5-VL and MedGemma, with In particular, for all critic free RL methods, the time for advantage calculation is less than 0.1% of the total training time. This reveals that the extra calculation in FairGRPO adds negligible runtime overhead.

### 4.4 RQ3: How does performance vary across individual demographic groups?

As shown in Fig. 3 and App. Tab. 4, FairGRPO demonstrates improved performance for both underrepresented and non-underrepresented groups. For example, in CheXpert, FairGRPO's F1 score is 24.4% higher for females and 34.4% higher for males compared to GRPO. Moreover, in PAD-UFES-20, FairGRPO improves performance by 6.33% on 75+ patients and 3.68% on patients aged 51-75 compared to GRPO. In addition, in the Hemorrhage dataset, FairGRPO improves performance by 18.70% on 51-75 group compared to GRPO. In CheXpert, our method also shows superiority for younger individuals (a1, a2), with an improvement of 31.45% on a1 and 24.32% on a2. These re-

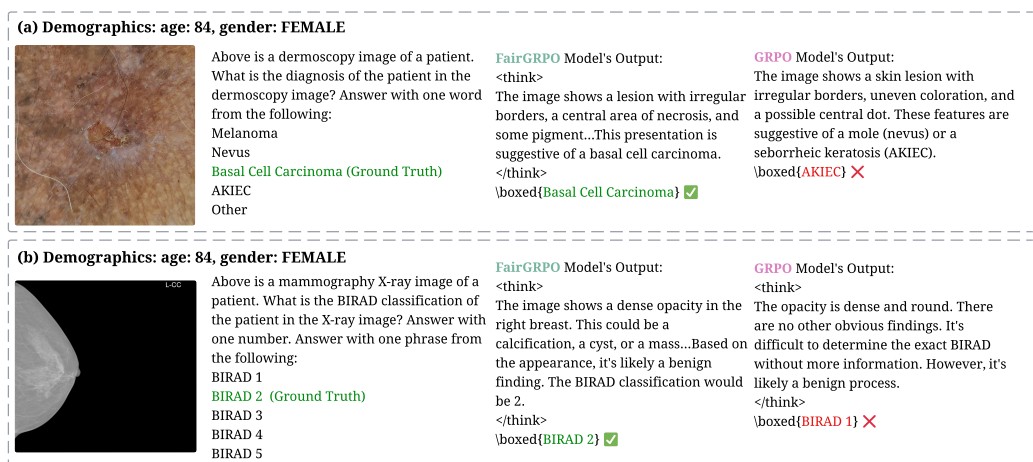

Figure 4: **Qualitative Examples of Model's Reasoning Traces.** We see the greatest performance boosts from underrepresented groups, including samples from older population and females. In particular, we observe the models trained with FairGRPO exhibit an accuracy improvement of 73.08% on 75+ populations in PAD-UFES-20 dataset, and a 36.53% on samples aged 51-75 in VinDr-Mammo. This figure shows examples of model's internal thinking process from the two groups.

sults demonstrate that our method provides consistent enhancements for elderly individuals across most datasets while showing minimal, if any, performance degradation for younger individuals, and in some cases even improvements. This indicates that the fairness improvements were not achieved at the expense of the majority group's performance.

## 4.5 QUALITATIVE ANALYSIS

Our qualitative analysis reveals that FairGRPO demonstrates superior diagnostic reasoning capabilities, particularly for underrepresented populations where GRPO exhibits increased hallucinations or unevidenced explanations. For example, in Fig. 4(a), examining an 84-year-old female's dermoscopy image, FairGRPO accurately identifies critical diagnostic features, including irregular borders, central necrosis, and distinctive pigmentation patterns, which leads to a correct Basal Cell Carcinoma diagnosis. Conversely, GRPO hallucinates non-existent features (a *central dot*), resulting in misdiagnosis of AKIEC. Similarly, Fig. 4(b) showcases FairGRPO's enhanced interpretive capability on another elderly female patient's mammography. FairMedGemma first identifies several possible diagnosis, including *a calcification, a cyst, or a mass*. It then correctly recognizes and contextualizes a dense opacity with rating BIRAD 2. GRPO trained model, on the other hand, underestimate the severity of the symptom, which results in a misclassification of BIRAD 1. These examples illustrate how FairGRPO's fairness-aware training not only improves quantitative metrics but also enhances the model's clinical reasoning quality, particularly benefiting historically underserved demographic groups.

## 5 CONCLUSION

In this work, we introduced FairGRPO, a novel reinforcement learning approach that addresses the challenge of demographic disparities in clinical AI systems. By implementing adaptive weighting based on demographics and task difficulty, FairGRPO ensures that minority and underrepresented groups receive equitable learning signals during training. Our evaluation across 7 clinical datasets demonstrates that FairGRPO not only reduces the disparities F1 scores across demographic groups by up to 28.9% but also improves overall model performance by 3.8% compared to vanilla GRPO. Through the release of FairMedGemma-4B, we provide the first publicly available clinical VLLM explicitly optimized for demographic fairness. Future works could explore extending FairGRPO to other medical modalities beyond vision-language tasks, and developing theoretical frameworks to better understand the convergence properties of fairness-aware RL. By establishing fairness as a fundamental optimization objective, we hope this work will inspire further research toward developing AI-assisted diagnostic systems that serve all patient populations equitably.

## 6 ETHICS STATEMENT

This work focuses on developing fairness-aware reinforcement learning methods for clinical diagnosis using vision-language models. We acknowledge the critical ethical considerations inherent in applying AI to healthcare and have taken careful steps to ensure our research adheres to the ethical standards.

All experiments in this study were conducted exclusively on publicly available, anonymized clinical datasets obtained in compliance with their respective licenses. Specifically, we used CheXpert, COVID-BLUES, VinDr-Mammo, ISIC-2020, HAM10000, PAD-UFES-20, and Hemorrhage datasets, each of which has been previously released for research purposes with appropriate de-identification procedures. No human subjects were directly involved in this research, and no new clinical data was collected. We do not redistribute these datasets; researchers interested in replicating our work should obtain them from the original sources in accordance with their respective terms of use.

Our work explicitly addresses demographic disparities in AI-assisted clinical diagnosis, recognizing that biased AI systems can perpetuate and amplify existing healthcare inequalities. By developing FairGRPO, we aim to reduce performance disparities across age and gender groups, thereby promoting more equitable healthcare AI. We acknowledge that fairness in healthcare is multifaceted and our demographic categorizations may not capture all relevant dimensions of patient diversity. Future work should consider additional protected attributes and intersectional identities.

While our methods demonstrate improved fairness metrics, we emphasize that these models are research prototypes and should not be used for actual clinical decision-making without proper regulatory approval and clinical validation. The deployment of AI in healthcare requires careful consideration of local regulations, clinical workflows, and continuous monitoring for unintended consequences.

## 7 REPRODUCIBILITY STATEMENT

To ensure reproducibility of the work, all experiments were conducted using publicly available datasets, which can be obtained from their respective original sources as detailed in Appendix B. Our complete training code, data preprocessing pipelines, and evaluation scripts are available at this anonymous link, while the trained model weights (FairMedGemma-4B) will be made available upon publication due to size constraints on the anonymous submission platform. All hyperparameters used in our experiments are comprehensively documented in Appendix A, including learning rates, batch sizes, rollout configurations, and training settings for both Qwen-2.5-VL and MedGemma models. We used the VERL framework for reinforcement learning implementation, with specific versions and dependencies listed in the repository's requirements file. Our fairness evaluation metrics are implemented with mathematical definitions provided in Appendix A.1.

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

# APPENDIX

## A  HYPERPARAMETERS & SETUPS

In this section, we describe our setup and hyperparameters during the training of the model. All models are trained with 4 NVIDIA H200 GPUs.

All experiments were conducted using the VERL (Volcano Engine Reinforcement Learning for LLMs) framework. The model was initialized from the pretrained MedGemma-4B-IT checkpoint and fine-tuned. We employed vLLM for efficient rollout generation with a GPU memory cache of 60% to balance between batch size and memory constraints. The relatively low learning rate of $5 \times 10^{-7}$ was chosen to ensure stable convergence given the complexity of the multi-task medical reasoning objective.

### A.1  EVALUATION METRICS

To comprehensively evaluate both performance and fairness across heterogeneous clinical subpopulations, we employ a hierarchical evaluation framework that prevents any single dataset or demographic subgroup from dominating the assessment.

**Notation.** Let $\mathcal{C}_k$ denote the set of classes for dataset $k$, and $\mathcal{G}$ denote the set of demographic groups. For each class $c \in \mathcal{C}_k$ and group $g \in \mathcal{G}$, we define: $TP_{c,g}$ (true positives), $FP_{c,g}$ (false positives), $TN_{c,g}$ (true negatives), and $FN_{c,g}$ (false negatives). Let $n_{c,g}$ denote the number of samples for class $c$ in group $g$.

**Performance Metrics.** We extract diagnoses from the model's free-text reasoning traces and evaluate each class as a binary classification problem. For class $c$ and group $g$:

$$\text{Acc}_{c,g} = \frac{TP_{c,g} + TN_{c,g}}{n_{c,g}}, \qquad \text{Precision}_{c,g} = \frac{TP_{c,g}}{TP_{c,g} + FP_{c,g}} \tag{3}$$

$$\text{Recall}_{c,g} = \frac{TP_{c,g}}{TP_{c,g} + FN_{c,g}}, \qquad \text{F1}_{c,g} = 2 \cdot \frac{\text{Precision}_{c,g} \cdot \text{Recall}_{c,g}}{\text{Precision}_{c,g} + \text{Recall}_{c,g}} \tag{4}$$

Table 3: Hyperparameters for All Trainings

| Parameter | Value |
|---|---|
| *Data Configuration* | |
| Train batch size | 512 |
| Validation batch size | 512 |
| Max prompt length | 4096 |
| Max response length | 4096 |
| *Model Configuration* | |
| Base model | MedGemma-4B-IT/Qwen2.5-VL-7B-Instruct |
| Tensor model parallel size | 2 |
| *Optimization* | |
| Learning rate | $5 \times 10^{-7}$ |
| PPO mini-batch size | 128 |
| PPO micro-batch size per GPU | 4 |
| KL | Disabled |
| *Rollout Configuration* | |
| Number of rollouts ($n$) | 10 |
| GPU memory utilization | 0.6 |
| Rollout engine | VLLM |
| *Training Settings* | |
| Total epochs | 15 |
| Validation frequency | 5 epochs |
| Model save frequency | 20 steps |
| Number of GPUs per node | 4 |
| Number of nodes | 1 |
| Critic warmup steps | 0 |

To ensure balanced representation across classes and datasets, we employ two-level averaging. For dataset $k$:

$$\mathrm{F1}_k = \frac{1}{|\mathcal{C}_k|} \sum_{c \in \mathcal{C}_k} \mathrm{F1}_c, \quad \text{where} \quad \mathrm{F1}_c = \frac{1}{|\mathcal{G}|} \sum_{g \in \mathcal{G}} \mathrm{F1}_{c,g} \tag{5}$$

The overall performance is then averaged across all $K$ datasets:

$$\mathrm{F1}_{\text{overall}} = \frac{1}{K} \sum_{k=1}^{K} \mathrm{F1}_k \tag{6}$$

This hierarchical averaging ensures that no single class or dataset dominates the final metrics, allowing the final metrics to be a balanced assessment across all 5 clinical domains.

**Fairness Metrics.** Following the popular approaches outlined in (Hort et al., 2024), we evaluate fairness through multiple complementary perspectives, each capturing different aspects of equitable model behavior across demographic groups. For each metric, we first compute dataset-level performance for each group, then assess disparities across groups.

*Equal Opportunity Difference (EOD):* We measure the disparity in true positive rates across groups to ensure equal diagnostic sensitivity:

$$\mathrm{EOD} = \max_{g \in \mathcal{G}} \mathrm{TPR}_g - \min_{g \in \mathcal{G}} \mathrm{TPR}_g, \quad \text{where} \quad \mathrm{TPR}_g = \frac{1}{K} \sum_{k=1}^{K} \frac{1}{|\mathcal{C}_k|} \sum_{c \in \mathcal{C}_k} \mathrm{TPR}_{c,g} \tag{7}$$

and $\mathrm{TPR}_{c,g} = \frac{TP_{c,g}}{TP_{c,g} + FN_{c,g}}$. A lower EOD indicates more equitable identification of positive cases, which is crucial for preventing delayed diagnoses in underserved populations.

*Predictive Parity:* We assess the reliability of positive predictions across groups through false discovery rate gaps:

$$\text{PP} = \max_{g \in \mathcal{G}} \text{FDR}_g - \min_{g \in \mathcal{G}} \text{FDR}_g, \quad \text{where} \quad \text{FDR}_g = \frac{1}{K} \sum_{k=1}^{K} \frac{1}{|\mathcal{C}_k|} \sum_{c \in \mathcal{C}_k} \text{FDR}_{c,g} \tag{8}$$

and $\text{FDR}_{c,g} = \frac{FP_{c,g}}{FP_{c,g} + TP_{c,g}}$. Lower predictive parity gaps ensure that positive predictions maintain consistent reliability across all demographic groups, fostering trust in AI-assisted diagnosis.

*False Positive Rate Difference:* We measure disparities in false positive rates to ensure equitable specificity across groups:

$$\text{FPR}_{\text{Diff}} = \max_{g \in \mathcal{G}} \text{FPR}_g - \min_{g \in \mathcal{G}} \text{FPR}_g \tag{9}$$

where $\text{FPR}_g$ follows the same hierarchical averaging structure as other group-level metrics. Lower FPR differences prevent differential overdiagnosis across demographic groups.

*Performance Disparities:* We directly measure accuracy and F1 score gaps to capture overall performance equity:

$$\Delta\text{Acc} = \max_{g \in \mathcal{G}} \text{Acc}_g - \min_{g \in \mathcal{G}} \text{Acc}_g, \qquad \Delta\text{F1} = \max_{g \in \mathcal{G}} \text{F1}_g - \min_{g \in \mathcal{G}} \text{F1}_g \tag{10}$$

where $\text{Acc}_g$ and $\text{F1}_g$ follow the same hierarchical averaging as $\text{TPR}_g$. Additionally, we compute the standard deviation of performance across groups to capture variability:

$$\sigma_{\text{Acc}} = \sqrt{\frac{1}{|\mathcal{G}|} \sum_{g \in \mathcal{G}} (\text{Acc}_g - \overline{\text{Acc}})^2}, \qquad \sigma_{\text{F1}} = \sqrt{\frac{1}{|\mathcal{G}|} \sum_{g \in \mathcal{G}} (\text{F1}_g - \overline{\text{F1}})^2} \tag{11}$$

where $\overline{\text{Acc}}$ and $\overline{\text{F1}}$ denote the mean values across all groups.

**Fairness-Utility Tradeoff.** To balance fairness and utility, we adopt Equity Scaling metrics following (Jin et al., 2024). These metrics combine performance with fairness considerations by penalizing models that achieve high average performance at the cost of large disparities across groups:

$$\text{Acc}_{\text{ES}} = \frac{\overline{\text{Acc}}}{1 + \sigma_{\text{Acc}}}, \qquad \text{F1}_{\text{ES}} = \frac{\overline{\text{F1}}}{1 + \sigma_{\text{F1}}} \tag{12}$$

These equity-scaled metrics reward models that achieve both high performance and low variance across demographic groups, providing a single scalar that captures the fairness-utility tradeoff. Higher values indicate better balance between overall performance and equitable distribution across all populations.

# B DATASET DETAILS

In this section, we provide a detailed description of datasets used in the experiments.

**CheXpert** (Irvin et al., 2019) is a public chest radiology dataset collected at Stanford Hospital, which contains 224,316 chest radiographs of 65,240 patients. Each record has an uncertain label of 14 diagnostic observations, including Atelectasis, Cardiomegaly, Consolidation, Edema, Enlarged Cardiomediastinum, Fracture, Lung Lesion, Lung Opacity, Pleural Effusion, Pneumonia, Pneumothorax, Pleural Other, Support Device and No Finding. We use a training set of 212,243 records, a test set of 225 records, and a total size of 212,498 records.

**COVID-BLUES** (Wiedemann et al., 2021) consists of bluepoint-specific lung ultrasound videos collected at the Maastricht University Medical Center in the Netherlands using the BLUE protocol. Each of the 63 patients has six recordings. Our evaluation focuses on two labels: the diagnostic label ("Has COVID", "No COVID"), and the patient age label. We use a training set of 266 records, a test set of 96 records, and a total size of 362 records.

**VinDr-Mammo** (Nguyen et al., 2021) contains mammography collected from Hospital 108 and Hanoi Medical University Hospital in Vietnam. The dataset includes local labels for bounding boxes; however, we evaluate our models based on the 5 global labels for BI-RADS 1-5. We use a training set of 16,000 records, a test set of 4,000 records, and a total size of 20,000 records.

**ISIC-2020** (Rotemberg et al., 2021) comprises dermoscopy of skin lesions from over 2,000 patients, generated by the International Skin Imaging Collaboration (ISIC). We evaluate the models on the binary classification ("Malignant" or "Benign") for each image, where all malignant diagnoses are histopathology–confirmed, while benign diagnoses are confirmed by expert agreement, longitudinal follow–up, or histopathology.We use a training set of 26,501 records, a test set of 6,625 records, and a total size of 33,126 records.

**HAM10000** (Tschandl et al., 2018a) is a dermoscopic image dataset released for the ISIC 2018 classification challenge, drawn from the ISIC archive. Our evaluation uses the diagnostic categories: Melanoma (MEL), Nevus (NV), Basal Cell Carcinoma (BCC), Actinic Keratosis/Intraepithelial Carcinoma (AKIEC), Other (OTHER).We use a training set of 8,012 records, a test set of 2,003 records, and a total size of 10,015 records.

**PAD-UFES-20** (Pacheco et al., 2020) comprises dermoscopy images of skin lesions with patient metadata collected at the Federal University of Espírito Santo by iPhone, which includes 1,641 skin lesions from 1,373 patients. We evaluate the models on the five skin diagnostics, three of which are skin disease and three of which are skin cancers: Melanoma (MEL), Nevus (NV), Basal Cell Carcinoma (BCC), Actinic Keratosis/Intraepithelial Carcinoma (AKIEC), Other (OTHER). All of the skin cancers are biopsy-proven, and more than half of the skin diseases are biopsy-proven as well. We use a training set of 1,839 records, a test set of 459 records, and a total size of 2,298 records.

**Hemorrhage** (Hssayeni et al., 2020) consists of intracranial hemorrhage CT images for 82 patients at Al Hilla Teaching Hospital, Iraq, each with brain and bone window images and approximately 30 image slices in total. We evaluate the models as binary diagnoses: "No Hemorrhage" and "Has Hemorrhage". We use a training set of 1,986 records, a test set of 515 patient records, and a total size of 2,501 records.

## C  THE USE OF LARGE LANGUAGE MODELS (LLMS)

We used ChatGPT for grammar corrections and debugging assistance, including explaining error messages and suggesting fixes. The model did not contribute research ideas, methods, experimental design, data, analyses or results. All changes were reviewed and implemented by the authors, who take full responsibility for the manuscript.

Table 4: **Relative F1 score improvements (%) for FairGRPO vs GRPO across demographic groups.** Values show the relative improvement ($\Delta\%$), GRPO baseline F1 score, and FairGRPO F1 score for each demographic group.

| Group | Dataset | | | | | |
|---|---|---|---|---|---|---|
| | **CheXpert** | **ISIC-2020** | **Hemorrhage** | **HAM10000** | **PAD-UFES-20** | **VinDr-Mammo** |
| **a1** | | | | | | |
| $\Delta\%$ | +31.44 | -0.14 | -3.40 | -20.95 | 0.00 | -6.21 |
| GRPO | 0.318 | 0.495 | 0.721 | 0.383 | 0.462 | 0.243 |
| FairGRPO | 0.418 | 0.494 | 0.696 | 0.302 | 0.462 | 0.228 |
| **a2** | | | | | | |
| $\Delta\%$ | +24.23 | -0.09 | +4.90 | -28.91 | +1.56 | +5.24 |
| GRPO | 0.296 | 0.496 | 0.600 | 0.262 | 0.385 | 0.234 |
| FairGRPO | 0.368 | 0.496 | 0.629 | 0.186 | 0.391 | 0.246 |
| **a3** | | | | | | |
| $\Delta\%$ | +33.18 | +1.65 | +18.77 | +39.18 | +3.71 | +11.29 |
| GRPO | 0.283 | 0.564 | 0.679 | 0.222 | 0.190 | 0.195 |
| FairGRPO | 0.377 | 0.574 | 0.806 | 0.309 | 0.197 | 0.217 |
| **a4** | | | | | | |
| $\Delta\%$ | +21.60 | +27.08 | – | +6.03 | +6.22 | -13.85 |
| GRPO | 0.302 | 0.469 | – | 0.185 | 0.221 | 0.238 |
| FairGRPO | 0.368 | 0.595 | – | 0.196 | 0.234 | 0.205 |
| **Female** | | | | | | |
| $\Delta\%$ | +24.45 | +6.90 | +3.54 | +19.21 | +4.20 | – |
| GRPO | 0.320 | 0.517 | 0.773 | 0.262 | 0.247 | – |
| FairGRPO | 0.398 | 0.553 | 0.800 | 0.313 | 0.258 | – |
| **Male** | | | | | | |
| $\Delta\%$ | +34.35 | +2.26 | +6.97 | +9.52 | +2.67 | – |
| GRPO | 0.253 | 0.546 | 0.628 | 0.240 | 0.214 | – |
| FairGRPO | 0.340 | 0.558 | 0.672 | 0.263 | 0.220 | – |
| **Average $\Delta\%$** | +28.21 | +6.28 | +6.16 | +4.01 | +3.06 | -0.88 |

Table 5: **Detailed fairness and performance metrics per dataset and demographic group for Reinforce++ on Qwen-2.5-VL.** Results shown for both age groups (a1-a4) and gender groups across all evaluation datasets. Higher values are better for accuracy, TPR, and F1; lower values are better for FPR and FDR.

| Dataset | Group | Performance Metrics | | | | Fairness Metrics | | | | Disparity Metrics | | | |
|---|---|---|---|---|---|---|---|---|---|---|---|---|---|
| | | Acc | F1 | TPR | FPR | FDR | $\sigma_{\text{Acc}}$ | $\sigma_{\text{F1}}$ | $\sigma_{\text{TPR}}$ | $\Delta$Acc | $\Delta$F1 | $\Delta$TPR | $\Delta$FPR |
| **Age Groups** | | | | | | | | | | | | | |
| **ChexPert** | a1 | .833 | .130 | .138 | .064 | .158 | .076 | .012 | .018 | .184 | .028 | .038 | .009 |
| | a2 | .748 | .102 | .118 | .068 | .139 | | | | | | | |
| | a3 | .770 | .120 | .114 | .070 | .202 | | | | | | | |
| | a4 | .649 | .125 | .151 | .074 | .223 | | | | | | | |
| **HAM10000** | a1 | .824 | .347 | .426 | .252 | .317 | .077 | .094 | .099 | .183 | .225 | .218 | .068 |
| | a2 | .876 | .200 | .231 | .197 | .759 | | | | | | | |
| | a3 | .783 | .239 | .262 | .185 | .669 | | | | | | | |
| | a4 | .693 | .122 | .208 | .197 | .660 | | | | | | | |
| **ISIC2020** | a1 | .979 | .595 | .595 | .405 | .405 | .071 | .045 | .043 | .157 | .100 | .099 | .099 |
| | a2 | .957 | .512 | .535 | .463 | .490 | | | | | | | |
| | a3 | .946 | .556 | .569 | .430 | .452 | | | | | | | |
| | a4 | .822 | .494 | .496 | .504 | .506 | | | | | | | |
| **PAD-UFES** | a1 | .813 | .417 | .357 | .000 | .000 | .033 | .076 | .062 | .081 | .161 | .145 | .195 |
| | a2 | .763 | .395 | .412 | .149 | .518 | | | | | | | |
| | a3 | .774 | .256 | .389 | .160 | .682 | | | | | | | |
| | a4 | .732 | .304 | .503 | .195 | .324 | | | | | | | |
| **Hemorrhage** | a1 | .728 | .444 | .445 | .555 | .557 | .048 | .059 | .062 | .093 | .116 | .120 | .120 |
| | a2 | .756 | .483 | .482 | .518 | .515 | | | | | | | |
| | a3 | .663 | .560 | .566 | .434 | .401 | | | | | | | |
| **VinDr** | a1 | .700 | .106 | .201 | .192 | .388 | .106 | .024 | .063 | .224 | .057 | .132 | .100 |
| | a2 | .709 | .132 | .204 | .196 | .573 | | | | | | | |
| | a3 | .724 | .162 | .225 | .189 | .563 | | | | | | | |
| | a4 | .500 | .121 | .333 | .289 | .593 | | | | | | | |
| **Gender Groups** | | | | | | | | | | | | | |
| **ChexPert** | Female | .716 | .123 | .129 | .072 | .129 | .046 | .006 | .011 | .065 | .009 | .016 | .009 |
| | Male | .781 | .115 | .112 | .063 | .183 | | | | | | | |
| **HAM10000** | Female | .842 | .230 | .249 | .187 | .709 | .021 | .001 | .002 | .030 | .001 | .003 | .003 |
| | Male | .812 | .231 | .246 | .190 | .689 | | | | | | | |
| **ISIC2020** | Female | .953 | .533 | .551 | .448 | .474 | .002 | .004 | .004 | .003 | .005 | .006 | .004 |
| | Male | .950 | .538 | .557 | .443 | .470 | | | | | | | |
| **PAD-UFES** | Female | .794 | .303 | .428 | .174 | .697 | .030 | .006 | .025 | .043 | .008 | .035 | .018 |
| | Male | .837 | .294 | .393 | .157 | .666 | | | | | | | |
| **Hemorrhage** | Female | .778 | .608 | .613 | .387 | .396 | .042 | .101 | .103 | .059 | .143 | .146 | .146 |
| | Male | .719 | .465 | .467 | .533 | .537 | | | | | | | |

Table 6: **Detailed fairness and performance metrics per dataset and demographic group for RLOO on Qwen-2.5-VL.** Results shown for both age groups (a1-a4) and gender groups across all evaluation datasets. Higher values are better for accuracy, TPR, and F1; lower values are better for FPR and FDR.

| Dataset | Group | Performance Metrics | | | | Fairness Metrics | | | | Disparity Metrics | | | |
|---------|-------|------|------|------|------|------|----------------|--------------|---------------|------|------|------|------|
| | | Acc | F1 | TPR | FPR | FDR | $\sigma_{Acc}$ | $\sigma_{F1}$ | $\sigma_{TPR}$ | $\Delta$Acc | $\Delta$F1 | $\Delta$TPR | $\Delta$FPR |
| **Age Groups** | | | | | | | | | | | | | |
| **ChexPert** | a1 | .833 | .285 | .338 | .096 | .221 | .066 | .067 | .083 | .161 | .149 | .186 | .080 |
| | a2 | .746 | .136 | .152 | .125 | .201 | | | | | | | |
| | a3 | .767 | .154 | .175 | .142 | .449 | | | | | | | |
| | a4 | .673 | .179 | .200 | .176 | .114 | | | | | | | |
| **HAM10000** | a1 | .924 | .314 | .327 | .309 | .364 | .114 | .087 | .051 | .243 | .199 | .107 | .116 |
| | a2 | .943 | .242 | .239 | .197 | .328 | | | | | | | |
| | a3 | .796 | .167 | .219 | .195 | .701 | | | | | | | |
| | a4 | .700 | .115 | .222 | .194 | .403 | | | | | | | |
| **ISIC2020** | a1 | .986 | .496 | .499 | .500 | .007 | .049 | .014 | .002 | .105 | .029 | .004 | .000 |
| | a2 | .990 | .497 | .500 | .500 | .005 | | | | | | | |
| | a3 | .974 | .493 | .499 | .500 | .013 | | | | | | | |
| | a4 | .886 | .468 | .495 | .500 | .056 | | | | | | | |
| **PAD-UFES** | a1 | .938 | .500 | .500 | .000 | .000 | .094 | .143 | .115 | .206 | .318 | .263 | .179 |
| | a2 | .760 | .371 | .410 | .172 | .544 | | | | | | | |
| | a3 | .764 | .233 | .309 | .155 | .623 | | | | | | | |
| | a4 | .732 | .182 | .237 | .179 | .716 | | | | | | | |
| **Hemorrhage** | a1 | .808 | .447 | .473 | .527 | .576 | .106 | .034 | .014 | .206 | .066 | .027 | .027 |
| | a2 | .869 | .465 | .494 | .506 | .561 | | | | | | | |
| | a3 | .663 | .399 | .500 | .500 | .169 | | | | | | | |
| **VinDr** | a1 | .807 | .137 | .200 | .200 | .096 | .094 | .036 | .066 | .211 | .086 | .133 | .136 |
| | a2 | .878 | .173 | .203 | .198 | .386 | | | | | | | |
| | a3 | .851 | .158 | .200 | .199 | .234 | | | | | | | |
| | a4 | .667 | .222 | .333 | .333 | .167 | | | | | | | |
| **Gender Groups** | | | | | | | | | | | | | |
| **ChexPert** | Female | .721 | .172 | .187 | .149 | .202 | .041 | .008 | .006 | .058 | .012 | .008 | .021 |
| | Male | .780 | .161 | .178 | .127 | .379 | | | | | | | |
| **HAM10000** | Female | .883 | .200 | .218 | .195 | .408 | .023 | .004 | .003 | .033 | .006 | .005 | .0003 |
| | Male | .850 | .194 | .223 | .194 | .781 | | | | | | | |
| **ISIC2020** | Female | .983 | .496 | .500 | .500 | .008 | .002 | .001 | .0004 | .003 | .001 | .001 | .000 |
| | Male | .980 | .495 | .499 | .500 | .009 | | | | | | | |
| **PAD-UFES** | Female | .788 | .265 | .387 | .172 | .653 | .025 | .024 | .037 | .036 | .034 | .052 | .019 |
| | Male | .823 | .230 | .335 | .153 | .680 | | | | | | | |
| **Hemorrhage** | Female | .821 | .451 | .490 | .510 | .583 | .005 | .001 | .001 | .006 | .002 | .002 | .002 |
| | Male | .814 | .449 | .488 | .512 | .585 | | | | | | | |

Table 7: **Detailed fairness and performance metrics per dataset and demographic group for GRPO on Qwen-2.5-VL.** Results shown for both age groups (a1-a4) and gender groups across all evaluation datasets. Higher values are better for accuracy, TPR, and F1; lower values are better for FPR and FDR.

| Dataset | Group | Performance Metrics | | | | Fairness Metrics | | | | Disparity Metrics | | | |
|---|---|---|---|---|---|---|---|---|---|---|---|---|---|
| | | Acc | F1 | TPR | FPR | FDR | $\sigma_{\mathbf{Acc}}$ | $\sigma_{\mathbf{F1}}$ | $\sigma_{\mathbf{TPR}}$ | $\Delta$**Acc** | $\Delta$**F1** | $\Delta$**TPR** | $\Delta$**FPR** |
| **Age Groups** | | | | | | | | | | | | | |
| **ChexPert** | a1 | .807 | .235 | .338 | .133 | .188 | .049 | .041 | .062 | .112 | .091 | .141 | .051 |
| | a2 | .766 | .192 | .228 | .136 | .160 | | | | | | | |
| | a3 | .785 | .163 | .196 | .141 | .160 | | | | | | | |
| | a4 | .695 | .254 | .282 | .184 | .096 | | | | | | | |
| **HAM10000** | a1 | .936 | .317 | .333 | .303 | .031 | .116 | .076 | .059 | .239 | .168 | .134 | .112 |
| | a2 | .943 | .185 | .199 | .194 | .427 | | | | | | | |
| | a3 | .796 | .170 | .223 | .192 | .411 | | | | | | | |
| | a4 | .703 | .149 | .242 | .191 | .266 | | | | | | | |
| **ISIC2020** | a1 | .987 | .497 | .500 | .500 | .007 | .048 | .013 | .000 | .101 | .027 | .000 | .000 |
| | a2 | .991 | .498 | .500 | .500 | .005 | | | | | | | |
| | a3 | .975 | .494 | .500 | .500 | .013 | | | | | | | |
| | a4 | .890 | .471 | .500 | .500 | .055 | | | | | | | |
| **PAD-UFES** | a1 | .875 | .917 | .857 | .000 | .000 | .048 | .320 | .270 | .104 | .716 | .597 | .167 |
| | a2 | .782 | .425 | .450 | .156 | .556 | | | | | | | |
| | a3 | .771 | .201 | .260 | .167 | .563 | | | | | | | |
| | a4 | .786 | .283 | .317 | .153 | .373 | | | | | | | |
| **Hemorrhage** | a1 | .854 | .461 | .500 | .500 | .073 | .119 | .038 | .000 | .217 | .069 | .000 | .000 |
| | a2 | .880 | .468 | .500 | .500 | .060 | | | | | | | |
| | a3 | .663 | .399 | .500 | .500 | .169 | | | | | | | |
| **VinDr** | a1 | .807 | .137 | .200 | .200 | .096 | .094 | .037 | .067 | .212 | .086 | .133 | .133 |
| | a2 | .879 | .164 | .200 | .200 | .061 | | | | | | | |
| | a3 | .852 | .155 | .200 | .200 | .074 | | | | | | | |
| | a4 | .667 | .222 | .333 | .333 | .167 | | | | | | | |
| **Gender Groups** | | | | | | | | | | | | | |
| **ChexPert** | Female | .742 | .220 | .250 | .160 | .154 | .036 | .036 | .037 | .051 | .051 | .053 | .031 |
| | Male | .793 | .169 | .198 | .129 | .125 | | | | | | | |
| **HAM10000** | Female | .882 | .194 | .216 | .191 | .589 | .021 | .010 | .012 | .029 | .015 | .018 | .001 |
| | Male | .852 | .208 | .234 | .190 | .537 | | | | | | | |
| **ISIC2020** | Female | .984 | .496 | .500 | .500 | .008 | .002 | .0004 | .000 | .003 | .001 | .000 | .000 |
| | Male | .981 | .495 | .500 | .500 | .009 | | | | | | | |
| **PAD-UFES** | Female | .800 | .308 | .374 | .174 | .451 | .026 | .038 | .044 | .037 | .054 | .063 | .016 |
| | Male | .837 | .255 | .311 | .158 | .539 | | | | | | | |
| **Hemorrhage** | Female | .838 | .456 | .500 | .500 | .081 | .002 | .001 | .000 | .003 | .001 | .000 | .000 |
| | Male | .834 | .455 | .500 | .500 | .083 | | | | | | | |

Table 8: **Detailed fairness and performance metrics per dataset and demographic group for GRPO with Resampling on Qwen-2.5-VL.** Results shown for both age groups (a1-a4) and gender groups across all evaluation datasets. Higher values are better for accuracy, TPR, and F1; lower values are better for FPR and FDR.

| Dataset | Group | Performance Metrics | | | | Fairness Metrics | | | | Disparity Metrics | | | |
|---|---|---|---|---|---|---|---|---|---|---|---|---|---|
| | | Acc | F1 | TPR | FPR | FDR | $\sigma_{Acc}$ | $\sigma_{F1}$ | $\sigma_{TPR}$ | $\Delta$Acc | $\Delta$F1 | $\Delta$TPR | $\Delta$FPR |
| **Age Groups** | | | | | | | | | | | | | |
| **ChexPert** | a1 | .847 | .142 | .150 | .056 | .153 | .083 | .018 | .029 | .203 | .044 | .059 | .022 |
| | a2 | .762 | .125 | .153 | .056 | .079 | | | | | | | |
| | a3 | .754 | .098 | .094 | .078 | .279 | | | | | | | |
| | a4 | .644 | .124 | .153 | .076 | .321 | | | | | | | |
| **HAM10000** | a1 | .785 | .307 | .354 | .302 | .341 | .057 | .071 | .070 | .138 | .158 | .160 | .111 |
| | a2 | .835 | .167 | .194 | .213 | .820 | | | | | | | |
| | a3 | .767 | .213 | .244 | .191 | .754 | | | | | | | |
| | a4 | .697 | .149 | .221 | .195 | .658 | | | | | | | |
| **ISIC2020** | a1 | .937 | .555 | .672 | .328 | .461 | .059 | .042 | .049 | .140 | .098 | .115 | .115 |
| | a2 | .894 | .490 | .557 | .443 | .494 | | | | | | | |
| | a3 | .885 | .522 | .585 | .415 | .477 | | | | | | | |
| | a4 | .797 | .588 | .616 | .384 | .422 | | | | | | | |
| **PAD-UFES** | a1 | .875 | .462 | .429 | .000 | .000 | .058 | .079 | .057 | .129 | .187 | .128 | .186 |
| | a2 | .769 | .408 | .406 | .148 | .557 | | | | | | | |
| | a3 | .771 | .275 | .390 | .162 | .673 | | | | | | | |
| | a4 | .746 | .372 | .518 | .186 | .535 | | | | | | | |
| **Hemorrhage** | a1 | .728 | .444 | .445 | .555 | .557 | .050 | .070 | .073 | .100 | .133 | .137 | .137 |
| | a2 | .785 | .471 | .472 | .528 | .530 | | | | | | | |
| | a3 | .685 | .577 | .582 | .418 | .361 | | | | | | | |
| **VinDr** | a1 | .696 | .168 | .283 | .192 | .531 | .056 | .029 | .065 | .116 | .062 | .146 | .010 |
| | a2 | .686 | .106 | .187 | .200 | .608 | | | | | | | |
| | a3 | .699 | .140 | .223 | .193 | .572 | | | | | | | |
| | a4 | .583 | .167 | .333 | .189 | .556 | | | | | | | |
| **Gender Groups** | | | | | | | | | | | | | |
| **ChexPert** | Female | .716 | .132 | .140 | .071 | .214 | .038 | .021 | .026 | .054 | .029 | .036 | .004 |
| | Male | .771 | .102 | .104 | .067 | .388 | | | | | | | |
| **HAM10000** | Female | .818 | .203 | .225 | .199 | .704 | .024 | .005 | .002 | .034 | .007 | .003 | .001 |
| | Male | .784 | .196 | .228 | .200 | .797 | | | | | | | |
| **ISIC2020** | Female | .901 | .512 | .581 | .419 | .484 | .013 | .00004 | .010 | .018 | .0001 | .014 | .014 |
| | Male | .882 | .512 | .595 | .405 | .482 | | | | | | | |
| **PAD-UFES** | Female | .800 | .338 | .493 | .168 | .706 | .014 | .014 | .076 | .020 | .019 | .107 | .003 |
| | Male | .820 | .318 | .386 | .165 | .678 | | | | | | | |
| **Hemorrhage** | Female | .803 | .572 | .564 | .436 | .405 | .048 | .064 | .057 | .067 | .091 | .081 | .081 |
| | Male | .736 | .481 | .484 | .516 | .519 | | | | | | | |

Table 9: **Detailed fairness and performance metrics per dataset and demographic group for GRPO with Group DRO on Qwen-2.5-VL.** Results shown for both age groups (a1-a4) and gender groups across all evaluation datasets. Higher values are better for accuracy, TPR, and F1; lower values are better for FPR and FDR.

| Dataset | Group | Performance Metrics | | | | Fairness Metrics | | | | Disparity Metrics | | | |
|---|---|---|---|---|---|---|---|---|---|---|---|---|---|
| | | Acc | F1 | TPR | FPR | FDR | $\sigma_{Acc}$ | $\sigma_{F1}$ | $\sigma_{TPR}$ | $\Delta$Acc | $\Delta$F1 | $\Delta$TPR | $\Delta$FPR |
| **Age Groups** | | | | | | | | | | | | | |
| **ChexPert** | a1 | .847 | .142 | .150 | .056 | .153 | | | | | | | |
| | a2 | .754 | .105 | .132 | .063 | .127 | .092 | .021 | .014 | .221 | .049 | .033 | .031 |
| | a3 | .767 | .115 | .117 | .070 | .124 | | | | | | | |
| | a4 | .625 | .092 | .132 | .087 | .245 | | | | | | | |
| **HAM10000** | a1 | .821 | .327 | .374 | .283 | .333 | | | | | | | |
| | a2 | .841 | .169 | .193 | .218 | .821 | .059 | .070 | .076 | .131 | .158 | .180 | .097 |
| | a3 | .769 | .239 | .257 | .191 | .629 | | | | | | | |
| | a4 | .710 | .190 | .249 | .186 | .498 | | | | | | | |
| **ISIC2020** | a1 | .953 | .579 | .680 | .320 | .445 | | | | | | | |
| | a2 | .923 | .501 | .554 | .446 | .492 | .072 | .044 | .084 | .164 | .101 | .203 | .203 |
| | a3 | .911 | .530 | .570 | .430 | .475 | | | | | | | |
| | a4 | .788 | .477 | .477 | .523 | .522 | | | | | | | |
| **PAD-UFES** | a1 | .875 | .462 | .429 | .000 | .000 | | | | | | | |
| | a2 | .760 | .399 | .397 | .155 | .545 | .062 | .082 | .056 | .139 | .188 | .125 | .195 |
| | a3 | .771 | .273 | .385 | .159 | .667 | | | | | | | |
| | a4 | .736 | .327 | .510 | .195 | .565 | | | | | | | |
| **Hemorrhage** | a1 | .748 | .452 | .457 | .543 | .551 | | | | | | | |
| | a2 | .840 | .478 | .490 | .510 | .523 | .089 | .041 | .047 | .177 | .080 | .092 | .092 |
| | a3 | .663 | .533 | .549 | .451 | .406 | | | | | | | |
| **VinDr** | a1 | .696 | .142 | .233 | .193 | .532 | | | | | | | |
| | a2 | .701 | .119 | .191 | .200 | .602 | .144 | .041 | .028 | .299 | .086 | .067 | .164 |
| | a3 | .716 | .141 | .210 | .192 | .577 | | | | | | | |
| | a4 | .417 | .056 | .167 | .356 | .633 | | | | | | | |
| **Gender Groups** | | | | | | | | | | | | | |
| **ChexPert** | Female | .715 | .118 | .136 | .071 | .130 | .040 | .010 | .019 | .057 | .014 | .026 | .003 |
| | Male | .772 | .105 | .110 | .067 | .195 | | | | | | | |
| **HAM10000** | Female | .825 | .258 | .258 | .196 | .613 | .026 | .030 | .013 | .037 | .042 | .019 | .002 |
| | Male | .788 | .216 | .239 | .198 | .725 | | | | | | | |
| **ISIC2020** | Female | .924 | .513 | .548 | .452 | .487 | .009 | .005 | .018 | .013 | .007 | .025 | .025 |
| | Male | .911 | .520 | .573 | .427 | .481 | | | | | | | |
| **PAD-UFES** | Female | .788 | .318 | .481 | .170 | .722 | .023 | .024 | .103 | .032 | .034 | .145 | .007 |
| | Male | .820 | .284 | .335 | .163 | .676 | | | | | | | |
| **Hemorrhage** | Female | .812 | .554 | .548 | .452 | .407 | .027 | .048 | .038 | .038 | .068 | .054 | .054 |
| | Male | .774 | .485 | .494 | .506 | .510 | | | | | | | |

Table 10: **Detailed fairness and performance metrics per dataset and demographic group for FairGRPO on Qwen-2.5-VL.** Results shown for both age groups (a1-a4) and gender groups across all evaluation datasets. Higher values are better for accuracy, TPR, and F1; lower values are better for FPR and FDR.

| Dataset | Group | Performance Metrics | | | | Fairness Metrics | | | | Disparity Metrics | | | |
|---|---|---|---|---|---|---|---|---|---|---|---|---|---|
| | | Acc | F1 | TPR | FPR | FDR | $\sigma_{Acc}$ | $\sigma_{F1}$ | $\sigma_{TPR}$ | $\Delta$Acc | $\Delta$F1 | $\Delta$TPR | $\Delta$FPR |
| **Age Groups** | | | | | | | | | | | | | |
| **ChexPert** | a1 | .813 | .161 | .225 | .109 | .140 | | | | | | | |
| | a2 | .771 | .149 | .166 | .096 | .104 | .063 | .015 | .030 | .142 | .031 | .065 | .068 |
| | a3 | .792 | .132 | .160 | .118 | .105 | | | | | | | |
| | a4 | .671 | .130 | .177 | .164 | .076 | | | | | | | |
| **HAM10000** | a1 | .915 | .304 | .300 | .212 | .024 | | | | | | | |
| | a2 | .920 | .209 | .464 | .148 | .606 | .089 | .045 | .075 | .183 | .096 | .164 | .069 |
| | a3 | .809 | .279 | .379 | .143 | .508 | | | | | | | |
| | a4 | .736 | .224 | .311 | .154 | .546 | | | | | | | |
| **ISIC2020** | a1 | .987 | .497 | .500 | .500 | .007 | | | | | | | |
| | a2 | .989 | .497 | .499 | .500 | .005 | .049 | .014 | .002 | .104 | .028 | .005 | .000 |
| | a3 | .972 | .492 | .497 | .500 | .013 | | | | | | | |
| | a4 | .886 | .468 | .495 | .500 | .056 | | | | | | | |
| **PAD-UFES** | a1 | .750 | .364 | .286 | .000 | .000 | | | | | | | |
| | a2 | .788 | .435 | .445 | .128 | .241 | .028 | .068 | .074 | .067 | .143 | .159 | .180 |
| | a3 | .817 | .292 | .294 | .139 | .301 | | | | | | | |
| | a4 | .779 | .291 | .317 | .180 | .205 | | | | | | | |
| **Hemorrhage** | a1 | .854 | .461 | .500 | .500 | .073 | | | | | | | |
| | a2 | .876 | .467 | .498 | .502 | .560 | .117 | .038 | .001 | .213 | .068 | .002 | .002 |
| | a3 | .663 | .399 | .500 | .500 | .169 | | | | | | | |
| **VinDr** | a1 | .807 | .137 | .200 | .200 | .096 | | | | | | | |
| | a2 | .879 | .164 | .200 | .200 | .061 | .094 | .037 | .067 | .212 | .086 | .133 | .134 |
| | a3 | .852 | .156 | .201 | .199 | .074 | | | | | | | |
| | a4 | .667 | .222 | .333 | .333 | .167 | | | | | | | |
| **Gender Groups** | | | | | | | | | | | | | |
| **ChexPert** | Female | .741 | .161 | .189 | .123 | .116 | .038 | .025 | .027 | .053 | .035 | .038 | .014 |
| | Male | .794 | .126 | .151 | .109 | .085 | | | | | | | |
| **HAM10000** | Female | .880 | .270 | .354 | .131 | .473 | .024 | .001 | .011 | .034 | .002 | .015 | .006 |
| | Male | .846 | .272 | .369 | .137 | .539 | | | | | | | |
| **ISIC2020** | Female | .982 | .495 | .498 | .500 | .008 | .002 | .0004 | .0001 | .002 | .001 | .0001 | .000 |
| | Male | .979 | .494 | .498 | .500 | .009 | | | | | | | |
| **PAD-UFES** | Female | .818 | .336 | .338 | .153 | .259 | .015 | .039 | .043 | .022 | .055 | .060 | .007 |
| | Male | .840 | .280 | .277 | .146 | .314 | | | | | | | |
| **Hemorrhage** | Female | .838 | .456 | .500 | .500 | .081 | .004 | .001 | .001 | .006 | .002 | .002 | .002 |
| | Male | .832 | .454 | .498 | .502 | .583 | | | | | | | |

Table 11: **Detailed fairness and performance metrics per dataset and demographic group for Reinforce++ on MedGemma.** Results shown for both age groups (a1-a4) and gender groups across all evaluation datasets. Higher values are better for accuracy, TPR, and F1; lower values are better for FPR and FDR.

| Dataset | Group | Performance Metrics | | | | Fairness Metrics | | | | Disparity Metrics | | | |
|---|---|---|---|---|---|---|---|---|---|---|---|---|---|
| | | Acc | F1 | TPR | FPR | FDR | $\sigma_{Acc}$ | $\sigma_{F1}$ | $\sigma_{TPR}$ | $\Delta$Acc | $\Delta$F1 | $\Delta$TPR | $\Delta$FPR |
| **Age Groups** | | | | | | | | | | | | | |
| **ChexPert** | a1 | .793 | .288 | .338 | .173 | .643 | .038 | .025 | .031 | .092 | .056 | .063 | .039 |
| | a2 | .745 | .260 | .299 | .161 | .659 | | | | | | | |
| | a3 | .761 | .269 | .298 | .161 | .647 | | | | | | | |
| | a4 | .702 | .316 | .361 | .134 | .483 | | | | | | | |
| **HAM10000** | a1 | .927 | .312 | .323 | .303 | .031 | .107 | .083 | .054 | .223 | .197 | .124 | .138 |
| | a2 | .938 | .233 | .233 | .165 | .474 | | | | | | | |
| | a3 | .801 | .183 | .223 | .178 | .586 | | | | | | | |
| | a4 | .716 | .115 | .199 | .167 | .402 | | | | | | | |
| **ISIC2020** | a1 | .987 | .497 | .500 | .500 | .007 | .048 | .017 | .005 | .101 | .042 | .010 | .010 |
| | a2 | .991 | .498 | .500 | .500 | .005 | | | | | | | |
| | a3 | .975 | .513 | .510 | .490 | .012 | | | | | | | |
| | a4 | .890 | .471 | .500 | .500 | .055 | | | | | | | |
| **PAD-UFES** | a1 | .875 | .462 | .429 | .000 | .000 | .056 | .122 | .095 | .118 | .253 | .192 | .159 |
| | a2 | .772 | .387 | .395 | .158 | .610 | | | | | | | |
| | a3 | .763 | .209 | .262 | .159 | .565 | | | | | | | |
| | a4 | .757 | .233 | .237 | .153 | .479 | | | | | | | |
| **Hemorrhage** | a1 | .871 | .731 | .643 | .208 | .087 | .066 | .118 | .066 | .124 | .234 | .126 | .057 |
| | a2 | .851 | .589 | .546 | .265 | .340 | | | | | | | |
| | a3 | .747 | .498 | .516 | .250 | .104 | | | | | | | |
| **VinDr** | a1 | .806 | .141 | .196 | .190 | .290 | .102 | .093 | .122 | .229 | .208 | .248 | .119 |
| | a2 | .867 | .186 | .204 | .196 | .592 | | | | | | | |
| | a3 | .836 | .177 | .200 | .199 | .620 | | | | | | | |
| | a4 | .639 | .349 | .444 | .308 | .708 | | | | | | | |
| **Gender Groups** | | | | | | | | | | | | | |
| **ChexPert** | Female | .757 | .332 | .358 | .134 | .561 | .004 | .068 | .067 | .006 | .097 | .094 | .032 |
| | Male | .751 | .235 | .264 | .166 | .687 | | | | | | | |
| **HAM10000** | Female | .879 | .202 | .216 | .180 | .555 | .016 | .005 | .006 | .023 | .008 | .008 | .012 |
| | Male | .856 | .210 | .224 | .168 | .538 | | | | | | | |
| **ISIC2020** | Female | .984 | .496 | .500 | .500 | .008 | .002 | .012 | .007 | .002 | .018 | .009 | .009 |
| | Male | .981 | .514 | .509 | .491 | .009 | | | | | | | |
| **PAD-UFES** | Female | .820 | .283 | .335 | .131 | .513 | .026 | .058 | .060 | .037 | .082 | .085 | .043 |
| | Male | .783 | .201 | .250 | .174 | .601 | | | | | | | |
| **Hemorrhage** | Female | .880 | .568 | .537 | .211 | .039 | .038 | .031 | .012 | .054 | .044 | .018 | .050 |
| | Male | .827 | .612 | .555 | .260 | .264 | | | | | | | |

Table 12: **Detailed fairness and performance metrics per dataset and demographic group for RLOO on MedGemma.** Results shown for both age groups (a1-a4) and gender groups across all evaluation datasets. Higher values are better for accuracy, TPR, and F1; lower values are better for FPR and FDR.

| Dataset | Group | Performance Metrics | | | | Fairness Metrics | | | | Disparity Metrics | | | |
|---|---|---|---|---|---|---|---|---|---|---|---|---|---|
| | | Acc | F1 | TPR | FPR | FDR | $\sigma_{Acc}$ | $\sigma_{F1}$ | $\sigma_{TPR}$ | $\Delta$Acc | $\Delta$F1 | $\Delta$TPR | $\Delta$FPR |
| **Age Groups** | | | | | | | | | | | | | |
| **ChexPert** | a1 | .900 | .533 | .600 | .082 | .185 | | | | | | | |
| | a2 | .817 | .363 | .429 | .110 | .241 | .077 | .091 | .090 | .189 | .193 | .196 | .100 |
| | a3 | .810 | .351 | .404 | .122 | .307 | | | | | | | |
| | a4 | .711 | .339 | .432 | .183 | .378 | | | | | | | |
| **HAM10000** | a1 | .933 | .316 | .333 | .333 | .033 | | | | | | | |
| | a2 | .938 | .183 | .195 | .195 | .628 | .110 | .065 | .058 | .228 | .139 | .138 | .147 |
| | a3 | .800 | .199 | .235 | .187 | .440 | | | | | | | |
| | a4 | .710 | .176 | .257 | .187 | .367 | | | | | | | |
| **ISIC2020** | a1 | .987 | .497 | .500 | .500 | .007 | | | | | | | |
| | a2 | .989 | .497 | .498 | .500 | .005 | .047 | .012 | .001 | .099 | .026 | .002 | .000 |
| | a3 | .974 | .493 | .499 | .500 | .013 | | | | | | | |
| | a4 | .890 | .471 | .500 | .500 | .055 | | | | | | | |
| **PAD-UFES** | a1 | .875 | .462 | .429 | .000 | .000 | | | | | | | |
| | a2 | .763 | .353 | .395 | .176 | .637 | .059 | .118 | .078 | .123 | .259 | .172 | .176 |
| | a3 | .752 | .203 | .316 | .174 | .582 | | | | | | | |
| | a4 | .757 | .234 | .257 | .162 | .453 | | | | | | | |
| **Hemorrhage** | a1 | .881 | .741 | .723 | .277 | .236 | | | | | | | |
| | a2 | .856 | .615 | .603 | .382 | .366 | .081 | .078 | .068 | .150 | .142 | .120 | .115 |
| | a3 | .730 | .598 | .608 | .392 | .204 | | | | | | | |
| **VinDr** | a1 | .807 | .138 | .200 | .197 | .294 | | | | | | | |
| | a2 | .878 | .167 | .200 | .198 | .587 | .067 | .152 | .151 | .155 | .319 | .303 | .081 |
| | a3 | .847 | .155 | .197 | .200 | .657 | | | | | | | |
| | a4 | .722 | .458 | .500 | .278 | .152 | | | | | | | |
| **Gender Groups** | | | | | | | | | | | | | |
| **ChexPert** | Female | .786 | .416 | .497 | .133 | .330 | .022 | .076 | .102 | .031 | .108 | .145 | .022 |
| | Male | .816 | .308 | .352 | .111 | .284 | | | | | | | |
| **HAM10000** | Female | .880 | .225 | .232 | .186 | .483 | .019 | .001 | .004 | .027 | .001 | .006 | .003 |
| | Male | .853 | .226 | .238 | .189 | .397 | | | | | | | |
| **ISIC2020** | Female | .982 | .495 | .498 | .500 | .008 | .002 | .0004 | .0001 | .002 | .001 | .0001 | .000 |
| | Male | .980 | .495 | .499 | .500 | .009 | | | | | | | |
| **PAD-UFES** | Female | .800 | .259 | .335 | .163 | .337 | .012 | .043 | .023 | .017 | .061 | .032 | .021 |
| | Male | .783 | .198 | .302 | .183 | .600 | | | | | | | |
| **Hemorrhage** | Female | .889 | .709 | .658 | .342 | .059 | .043 | .049 | .025 | .061 | .069 | .035 | .027 |
| | Male | .828 | .639 | .623 | .369 | .329 | | | | | | | |

Table 13: **Detailed fairness and performance metrics per dataset and demographic group for GRPO on MedGemma.** Results shown for both age groups (a1-a4) and gender groups across all evaluation datasets. Higher values are better for accuracy, TPR, and F1; lower values are better for FPR and FDR.

| Dataset | Group | Performance Metrics | | | | Fairness Metrics | | | | Disparity Metrics | | | |
|---|---|---|---|---|---|---|---|---|---|---|---|---|---|
| | | Acc | F1 | TPR | FPR | FDR | $\sigma_{\text{Acc}}$ | $\sigma_{\text{F1}}$ | $\sigma_{\text{TPR}}$ | $\Delta$Acc | $\Delta$F1 | $\Delta$TPR | $\Delta$FPR |
| **Age Groups** | | | | | | | | | | | | | |
| **ChexPert** | a1 | .893 | .318 | .342 | .038 | .092 | .084 | .015 | .040 | .202 | .035 | .093 | .050 |
| | a2 | .814 | .296 | .279 | .059 | .174 | | | | | | | |
| | a3 | .824 | .283 | .248 | .052 | .159 | | | | | | | |
| | a4 | .691 | .302 | .306 | .088 | .211 | | | | | | | |
| **HAM10000** | a1 | .918 | .383 | .369 | .279 | .252 | .108 | .086 | .057 | .233 | .198 | .120 | .103 |
| | a2 | .943 | .262 | .248 | .177 | .425 | | | | | | | |
| | a3 | .802 | .222 | .249 | .187 | .553 | | | | | | | |
| | a4 | .710 | .185 | .271 | .185 | .498 | | | | | | | |
| **ISIC2020** | a1 | .983 | .495 | .496 | .500 | .007 | .048 | .041 | .020 | .102 | .096 | .040 | .039 |
| | a2 | .988 | .496 | .497 | .500 | .505 | | | | | | | |
| | a3 | .974 | .564 | .536 | .462 | .012 | | | | | | | |
| | a4 | .886 | .468 | .495 | .500 | .056 | | | | | | | |
| **PAD-UFES** | a1 | .875 | .462 | .429 | .000 | .000 | .059 | .130 | .092 | .125 | .272 | .179 | .179 |
| | a2 | .779 | .385 | .421 | .163 | .600 | | | | | | | |
| | a3 | .751 | .190 | .287 | .179 | .598 | | | | | | | |
| | a4 | .750 | .220 | .249 | .171 | .230 | | | | | | | |
| **Hemorrhage** | a1 | .858 | .721 | .692 | .236 | .247 | .036 | .062 | .057 | .071 | .121 | .113 | .105 |
| | a2 | .836 | .600 | .579 | .340 | .376 | | | | | | | |
| | a3 | .787 | .679 | .650 | .259 | .150 | | | | | | | |
| **VinDr** | a1 | .804 | .243 | .288 | .177 | .553 | .064 | .022 | .031 | .146 | .048 | .069 | .023 |
| | a2 | .841 | .234 | .267 | .189 | .764 | | | | | | | |
| | a3 | .808 | .195 | .219 | .188 | .796 | | | | | | | |
| | a4 | .694 | .238 | .278 | .200 | .458 | | | | | | | |
| **Gender Groups** | | | | | | | | | | | | | |
| **ChexPert** | Female | .779 | .320 | .290 | .071 | .281 | .032 | .047 | .042 | .045 | .067 | .059 | .020 |
| | Male | .824 | .253 | .231 | .051 | .234 | | | | | | | |
| **HAM10000** | Female | .885 | .262 | .261 | .178 | .386 | .022 | .015 | .008 | .032 | .022 | .012 | .008 |
| | Male | .854 | .240 | .249 | .186 | .593 | | | | | | | |
| **ISIC2020** | Female | .982 | .517 | .509 | .489 | .008 | .002 | .020 | .011 | .003 | .029 | .016 | .016 |
| | Male | .979 | .546 | .525 | .472 | .134 | | | | | | | |
| **PAD-UFES** | Female | .797 | .247 | .325 | .163 | .533 | .003 | .023 | .006 | .004 | .033 | .009 | .020 |
| | Male | .793 | .214 | .316 | .184 | .392 | | | | | | | |
| **Hemorrhage** | Female | .902 | .773 | .722 | .221 | .130 | .062 | .102 | .089 | .088 | .144 | .126 | .090 |
| | Male | .814 | .628 | .596 | .310 | .327 | | | | | | | |

Table 14: **Detailed fairness and performance metrics per dataset and demographic group for GRPO with Resampling on MedGemma.** Results shown for both age groups (a1-a4) and gender groups across all evaluation datasets. Higher values are better for accuracy, TPR, and F1; lower values are better for FPR and FDR.

| Dataset | Group | Performance Metrics | | | | Fairness Metrics | | | | Disparity Metrics | | | |
|---|---|---|---|---|---|---|---|---|---|---|---|---|---|
| | | Acc | F1 | TPR | FPR | FDR | $\sigma_{Acc}$ | $\sigma_{F1}$ | $\sigma_{TPR}$ | $\Delta$Acc | $\Delta$F1 | $\Delta$TPR | $\Delta$FPR |
| **Age Groups** | | | | | | | | | | | | | |
| **ChexPert** | a1 | .913 | .466 | .500 | .045 | .061 | | | | | | | |
| | a2 | .832 | .349 | .389 | .072 | .202 | .072 | .066 | .066 | .175 | .147 | .157 | .061 |
| | a3 | .828 | .319 | .343 | .074 | .284 | | | | | | | |
| | a4 | .738 | .343 | .400 | .106 | .272 | | | | | | | |
| **HAM10000** | a1 | .942 | .320 | .333 | .242 | .025 | | | | | | | |
| | a2 | .922 | .187 | .387 | .181 | .222 | .111 | .074 | .037 | .236 | .170 | .084 | .070 |
| | a3 | .794 | .200 | .303 | .173 | .248 | | | | | | | |
| | a4 | .707 | .151 | .312 | .181 | .500 | | | | | | | |
| **ISIC2020** | a1 | .987 | .497 | .500 | .500 | .007 | | | | | | | |
| | a2 | .991 | .498 | .500 | .500 | .005 | .048 | .013 | .000 | .101 | .027 | .000 | .000 |
| | a3 | .975 | .494 | .500 | .500 | .013 | | | | | | | |
| | a4 | .890 | .471 | .500 | .500 | .055 | | | | | | | |
| **PAD-UFES** | a1 | .813 | .417 | .357 | .000 | .000 | | | | | | | |
| | a2 | .821 | .397 | .449 | .137 | .163 | .024 | .115 | .085 | .050 | .237 | .187 | .184 |
| | a3 | .771 | .180 | .261 | .179 | .355 | | | | | | | |
| | a4 | .783 | .250 | .281 | .184 | .364 | | | | | | | |
| **Hemorrhage** | a1 | .828 | .453 | .484 | .516 | .575 | | | | | | | |
| | a2 | .873 | .576 | .561 | .439 | .345 | .110 | .091 | .041 | .210 | .178 | .077 | .077 |
| | a3 | .663 | .399 | .500 | .500 | .169 | | | | | | | |
| **VinDr** | a1 | .807 | .172 | .221 | .193 | .443 | | | | | | | |
| | a2 | .871 | .184 | .241 | .197 | .622 | .064 | .035 | .051 | .149 | .078 | .112 | .030 |
| | a3 | .840 | .195 | .234 | .195 | .626 | | | | | | | |
| | a4 | .722 | .250 | .333 | .222 | .133 | | | | | | | |
| **Gender Groups** | | | | | | | | | | | | | |
| **ChexPert** | Female | .810 | .410 | .435 | .068 | .198 | .015 | .086 | .085 | .021 | .121 | .121 | .007 |
| | Male | .831 | .288 | .315 | .075 | .203 | | | | | | | |
| **HAM10000** | Female | .869 | .199 | .288 | .164 | .430 | .017 | .015 | .020 | .025 | .021 | .028 | .006 |
| | Male | .845 | .220 | .316 | .170 | .223 | | | | | | | |
| **ISIC2020** | Female | .984 | .496 | .500 | .500 | .008 | .002 | .0004 | .000 | .003 | .001 | .000 | .000 |
| | Male | .981 | .495 | .500 | .500 | .009 | | | | | | | |
| **PAD-UFES** | Female | .775 | .238 | .336 | .178 | .277 | .027 | .040 | .025 | .039 | .057 | .035 | .002 |
| | Male | .737 | .181 | .301 | .180 | .544 | | | | | | | |
| **Hemorrhage** | Female | .838 | .456 | .500 | .500 | .081 | .013 | .039 | .015 | .018 | .055 | .021 | .021 |
| | Male | .819 | .511 | .521 | .479 | .424 | | | | | | | |

Table 15: **Detailed fairness and performance metrics per dataset and demographic group for GRPO with Group DRO on MedGemma.** Results shown for both age groups (a1-a4) and gender groups across all evaluation datasets. Higher values are better for accuracy, TPR, and F1; lower values are better for FPR and FDR.

| Dataset | Group | Performance Metrics | | | | Fairness Metrics | | | | Disparity Metrics | | | |
|---|---|---|---|---|---|---|---|---|---|---|---|---|---|
| | | Acc | F1 | TPR | FPR | FDR | $\sigma_{Acc}$ | $\sigma_{F1}$ | $\sigma_{TPR}$ | $\Delta$Acc | $\Delta$F1 | $\Delta$TPR | $\Delta$FPR |
| **Age Groups** | | | | | | | | | | | | | |
| **ChexPert** | a1 | .913 | .380 | .400 | .030 | .136 | .064 | .029 | .032 | .157 | .060 | .070 | .062 |
| | a2 | .848 | .390 | .402 | .049 | .168 | | | | | | | |
| | a3 | .836 | .330 | .341 | .065 | .228 | | | | | | | |
| | a4 | .756 | .390 | .411 | .093 | .225 | | | | | | | |
| **HAM10000** | a1 | .945 | .509 | .467 | .273 | .028 | .116 | .162 | .116 | .245 | .373 | .246 | .088 |
| | a2 | .930 | .239 | .241 | .194 | .559 | | | | | | | |
| | a3 | .801 | .215 | .245 | .184 | .512 | | | | | | | |
| | a4 | .700 | .136 | .221 | .194 | .374 | | | | | | | |
| **ISIC2020** | a1 | .987 | .497 | .500 | .500 | .007 | .048 | .013 | .0002 | .100 | .027 | .0003 | .000 |
| | a2 | .990 | .497 | .500 | .500 | .005 | | | | | | | |
| | a3 | .975 | .494 | .500 | .500 | .013 | | | | | | | |
| | a4 | .890 | .471 | .500 | .500 | .055 | | | | | | | |
| **PAD-UFES** | a1 | .875 | .462 | .429 | .000 | .000 | .065 | .149 | .104 | .154 | .315 | .223 | .196 |
| | a2 | .782 | .354 | .454 | .149 | .704 | | | | | | | |
| | a3 | .764 | .176 | .309 | .175 | .434 | | | | | | | |
| | a4 | .721 | .147 | .231 | .196 | .142 | | | | | | | |
| **Hemorrhage** | a1 | .861 | .735 | .749 | .251 | .276 | .030 | .035 | .032 | .053 | .066 | .063 | .063 |
| | a2 | .862 | .681 | .686 | .314 | .323 | | | | | | | |
| | a3 | .809 | .747 | .725 | .275 | .141 | | | | | | | |
| **VinDr** | a1 | .811 | .175 | .225 | .192 | .227 | .080 | .029 | .054 | .182 | .063 | .115 | .085 |
| | a2 | .876 | .173 | .218 | .198 | .255 | | | | | | | |
| | a3 | .847 | .186 | .233 | .196 | .246 | | | | | | | |
| | a4 | .694 | .235 | .333 | .278 | .152 | | | | | | | |
| **Gender Groups** | | | | | | | | | | | | | |
| **ChexPert** | Female | .815 | .397 | .411 | .064 | .226 | .021 | .059 | .058 | .030 | .083 | .082 | .008 |
| | Male | .845 | .315 | .328 | .055 | .180 | | | | | | | |
| **HAM10000** | Female | .878 | .245 | .245 | .181 | .500 | .020 | .015 | .006 | .028 | .021 | .008 | .004 |
| | Male | .850 | .224 | .237 | .186 | .502 | | | | | | | |
| **ISIC2020** | Female | .983 | .496 | .500 | .500 | .008 | .002 | .0003 | .0003 | .002 | .0004 | .0004 | .000 |
| | Male | .981 | .495 | .500 | .500 | .009 | | | | | | | |
| **PAD-UFES** | Female | .800 | .208 | .321 | .178 | .443 | .021 | .005 | .007 | .030 | .007 | .010 | .007 |
| | Male | .830 | .201 | .331 | .171 | .238 | | | | | | | |
| **Hemorrhage** | Female | .923 | .833 | .784 | .216 | .080 | .065 | .101 | .069 | .091 | .143 | .098 | .098 |
| | Male | .832 | .690 | .687 | .313 | .306 | | | | | | | |

Table 16: **Detailed fairness and performance metrics per dataset and demographic group for FairGRPO$_{ND}$ on MedGemma.** Results shown for both age groups (a1-a4) and gender groups across all evaluation datasets. Higher values are better for accuracy, TPR, and F1; lower values are better for FPR and FDR.

| Dataset | Group | Performance Metrics | | | | Fairness Metrics | | | | Disparity Metrics | | | |
|---|---|---|---|---|---|---|---|---|---|---|---|---|---|
| | | Acc | F1 | TPR | FPR | FDR | $\sigma_{Acc}$ | $\sigma_{F1}$ | $\sigma_{TPR}$ | $\Delta$Acc | $\Delta$F1 | $\Delta$TPR | $\Delta$FPR |
| **Age Groups** | | | | | | | | | | | | | |
| **ChexPert** | a1 | .860 | .342 | .475 | .096 | .207 | | | | | | | |
| | a2 | .800 | .366 | .387 | .146 | .523 | .057 | .037 | .044 | .138 | .087 | .088 | .101 |
| | a3 | .799 | .379 | .390 | .162 | .544 | | | | | | | |
| | a4 | .722 | .430 | .449 | .198 | .393 | | | | | | | |
| **HAM10000** | a1 | .897 | .301 | .296 | .255 | .694 | | | | | | | |
| | a2 | .905 | .260 | .225 | .151 | .597 | .077 | .035 | .029 | .163 | .084 | .071 | .121 |
| | a3 | .814 | .270 | .264 | .134 | .528 | | | | | | | |
| | a4 | .741 | .216 | .255 | .144 | .635 | | | | | | | |
| **ISIC2020** | a1 | .980 | .493 | .493 | .500 | .007 | | | | | | | |
| | a2 | .988 | .496 | .497 | .500 | .505 | .048 | .039 | .020 | .102 | .091 | .042 | .038 |
| | a3 | .973 | .559 | .535 | .463 | .262 | | | | | | | |
| | a4 | .886 | .468 | .495 | .500 | .056 | | | | | | | |
| **PAD-UFES** | a1 | .938 | .500 | .500 | .000 | .000 | | | | | | | |
| | a2 | .795 | .408 | .424 | .142 | .580 | .078 | .117 | .103 | .174 | .279 | .235 | .152 |
| | a3 | .764 | .221 | .265 | .152 | .747 | | | | | | | |
| | a4 | .797 | .352 | .333 | .137 | .612 | | | | | | | |
| **Hemorrhage** | a1 | .821 | .694 | .722 | .270 | .321 | | | | | | | |
| | a2 | .835 | .655 | .665 | .324 | .352 | .016 | .046 | .033 | .031 | .092 | .059 | .066 |
| | a3 | .803 | .747 | .725 | .259 | .183 | | | | | | | |
| **VinDr** | a1 | .794 | .191 | .219 | .181 | .429 | | | | | | | |
| | a2 | .820 | .198 | .199 | .191 | .794 | .043 | .133 | .202 | .098 | .270 | .412 | .052 |
| | a3 | .800 | .196 | .201 | .187 | .606 | | | | | | | |
| | a4 | .722 | .460 | .611 | .233 | .292 | | | | | | | |
| **Gender Groups** | | | | | | | | | | | | | |
| **ChexPert** | Female | .767 | .397 | .413 | .180 | .513 | .032 | .026 | .037 | .046 | .037 | .053 | .042 |
| | Male | .813 | .360 | .360 | .138 | .541 | | | | | | | |
| **HAM10000** | Female | .871 | .279 | .278 | .132 | .509 | .019 | .017 | .024 | .027 | .024 | .034 | .004 |
| | Male | .845 | .255 | .244 | .136 | .588 | | | | | | | |
| **ISIC2020** | Female | .980 | .515 | .508 | .489 | .408 | .0005 | .021 | .012 | .001 | .030 | .017 | .017 |
| | Male | .979 | .545 | .525 | .473 | .209 | | | | | | | |
| **PAD-UFES** | Female | .823 | .306 | .377 | .122 | .685 | .020 | .054 | .044 | .028 | .076 | .062 | .038 |
| | Male | .795 | .231 | .315 | .159 | .569 | | | | | | | |
| **Hemorrhage** | Female | .906 | .815 | .795 | .205 | .160 | .074 | .109 | .094 | .104 | .154 | .133 | .115 |
| | Male | .802 | .662 | .663 | .319 | .338 | | | | | | | |

Table 17: **Detailed fairness and performance metrics per dataset and demographic group for FairGRPO on MedGemma.** Results shown for both age groups (a1-a4) and gender groups across all evaluation datasets. Higher values are better for accuracy, TPR, and F1; lower values are better for FPR and FDR.

| Dataset | Group | Performance Metrics | | | | Fairness Metrics | | | | Disparity Metrics | | | |
|---|---|---|---|---|---|---|---|---|---|---|---|---|---|
| | | Acc | F1 | TPR | FPR | FDR | $\sigma_{Acc}$ | $\sigma_{F1}$ | $\sigma_{TPR}$ | $\Delta$Acc | $\Delta$F1 | $\Delta$TPR | $\Delta$FPR |
| **Age Groups** | | | | | | | | | | | | | |
| **ChexPert** | a1 | .900 | .359 | .388 | .045 | .063 | | | | | | | |
| | a2 | .828 | .354 | .351 | .063 | .224 | .062 | .019 | .035 | .151 | .047 | .076 | .056 |
| | a3 | .833 | .328 | .330 | .065 | .239 | | | | | | | |
| | a4 | .749 | .375 | .406 | .101 | .332 | | | | | | | |
| **HAM10000** | a1 | .933 | .315 | .327 | .273 | .028 | | | | | | | |
| | a2 | .941 | .251 | .238 | .191 | .227 | .114 | .074 | .043 | .238 | .171 | .088 | .083 |
| | a3 | .799 | .196 | .241 | .191 | .236 | | | | | | | |
| | a4 | .703 | .144 | .242 | .190 | .312 | | | | | | | |
| **ISIC2020** | a1 | .987 | .497 | .500 | .500 | .007 | | | | | | | |
| | a2 | .991 | .498 | .500 | .500 | .005 | .048 | .013 | .000 | .101 | .027 | .000 | .000 |
| | a3 | .975 | .494 | .500 | .500 | .013 | | | | | | | |
| | a4 | .890 | .471 | .500 | .500 | .055 | | | | | | | |
| **PAD-UFES** | a1 | .875 | .462 | .429 | .000 | .000 | | | | | | | |
| | a2 | .846 | .507 | .515 | .118 | .214 | .034 | .111 | .092 | .082 | .218 | .211 | .164 |
| | a3 | .825 | .289 | .351 | .128 | .311 | | | | | | | |
| | a4 | .793 | .299 | .304 | .164 | .203 | | | | | | | |
| **Hemorrhage** | a1 | .854 | .728 | .745 | .255 | .286 | | | | | | | |
| | a2 | .840 | .631 | .634 | .366 | .372 | .023 | .062 | .059 | .045 | .116 | .111 | .111 |
| | a3 | .809 | .747 | .725 | .275 | .141 | | | | | | | |
| **VinDr** | a1 | .807 | .137 | .200 | .200 | .096 | | | | | | | |
| | a2 | .879 | .164 | .200 | .200 | .061 | .094 | .037 | .067 | .212 | .086 | .133 | .133 |
| | a3 | .852 | .155 | .200 | .200 | .074 | | | | | | | |
| | a4 | .667 | .222 | .333 | .333 | .167 | | | | | | | |
| **Gender Groups** | | | | | | | | | | | | | |
| **ChexPert** | Female | .810 | .399 | .406 | .070 | .297 | .018 | .075 | .080 | .026 | .106 | .113 | .010 |
| | Male | .835 | .293 | .292 | .060 | .214 | | | | | | | |
| **HAM10000** | Female | .883 | .240 | .254 | .187 | .223 | .022 | .021 | .013 | .032 | .030 | .018 | .004 |
| | Male | .851 | .211 | .236 | .192 | .229 | | | | | | | |
| **ISIC2020** | Female | .984 | .496 | .500 | .500 | .008 | .002 | .0004 | .000 | .003 | .001 | .000 | .000 |
| | Male | .981 | .495 | .500 | .500 | .009 | | | | | | | |
| **PAD-UFES** | Female | .831 | .328 | .384 | .138 | .286 | .014 | .030 | .002 | .019 | .042 | .003 | .006 |
| | Male | .812 | .286 | .387 | .144 | .325 | | | | | | | |
| **Hemorrhage** | Female | .889 | .758 | .722 | .278 | .173 | .046 | .057 | .032 | .065 | .080 | .045 | .046 |
| | Male | .824 | .678 | .676 | .324 | .320 | | | | | | | |

Table 18: **RQ1: Fairness and performance metrics comparison against RL and fairness mitigation baselines.** For fairness metrics, lower values are better and are indicated by ↓. For performance and combined metrics, higher values are better and are indicated by ↑. Bold values indicate the best result in each column for each model separately. **FairGRPO$_{ND}$** is the ablation of **FairGRPO** where the model does not have access to the ground truth demographic information, and the groups are inferred entirely via clustering. We release **MedGemma** trained with **FairGRPO** as **FairMedGemma**. Results show mean ± std over 4 training runs. Detailed per dataset metrics are included in App. Tab. 5-17.

| Training Method | Fairness Metrics | | | | | | |
|---|---|---|---|---|---|---|---|
| | PP ↓ | EOD ↓ | FPR$_{Diff}$ ↓ | $\sigma_{F1}$ ↓ | $\Delta$F1 ↓ | $\sigma_{Acc}$ ↓ | $\Delta$Acc ↓ |
| **Qwen-2.5-VL-7B** | | | | | | | |
| Re++ (Hu, 2025) | 16.66 ± 2.11 | 6.66 ± 1.59 | 6.37 ± 0.20 | .0322 ± .0000 | .0647 ± .0004 | 5.06 ± 0.49 | 10.33 ± 1.01 |
| RLOO (Ahmadian et al., 2024) | 22.34 ± 0.86 | 6.67 ± 0.13 | 5.68 ± 0.80 | .0330 ± .0006 | .0693 ± .0017 | 4.86 ± 0.33 | 10.00 ± 0.79 |
| GRPO (Shao et al., 2024) | 17.90 ± 9.21 | 7.93 ± 1.64 | **4.85 ± 0.34** | .0387 ± .0107 | .0821 ± .0215 | 4.85 ± 0.24 | 9.92 ± 0.69 |
| GRPO+RS (Puyol-Antón et al., 2021) | 19.62 ± 7.22 | 6.85 ± 0.80 | 6.44 ± 1.39 | .0319 ± .0009 | .0628 ± .0037 | 5.50 ± 0.17 | 11.26 ± 0.34 |
| **FairGRPO** | **15.42 ± 1.95** | **5.62 ± 0.10** | 5.00 ± 0.87 | **.0254 ± .0035** | **.0522 ± .0099** | **4.42 ± 0.01** | **8.95 ± 0.03** |
| **MedGemma-4B** | | | | | | | |
| Re++ (Hu, 2025) | 20.30 ± 0.97 | 7.78 ± 1.37 | 5.69 ± 0.10 | .0469 ± .0069 | .0898 ± .0191 | 4.44 ± 0.17 | 8.99 ± 0.25 |
| RLOO (Ahmadian et al., 2024) | 20.45 ± 4.57 | 10.35 ± 0.03 | 5.51 ± 0.01 | .0592 ± .0011 | .1173 ± .0004 | 4.29 ± 0.07 | 8.79 ± 0.06 |
| GRPO (Shao et al., 2024) | 20.89 ± 2.16 | **6.30 ± 0.25** | 5.26 ± 0.62 | .0387 ± .0045 | .0753 ± .0059 | 4.19 ± 0.03 | 8.57 ± 0.03 |
| GRPO+RS (Puyol-Antón et al., 2021) | 24.55 ± 1.12 | 6.97 ± 0.44 | **4.78 ± 1.84** | .0422 ± .0047 | .0834 ± .0093 | 4.20 ± 0.21 | 8.77 ± 0.54 |
| GRPO+DRO (Sagawa et al., 2019) | 18.20 ± 3.06 | 7.52 ± 0.22 | 5.68 ± 0.98 | .0456 ± .0013 | .0895 ± .0034 | 4.55 ± 0.26 | 9.39 ± 0.61 |
| **FairGRPO$_{ND}$** | 24.87 ± 0.40 | 9.09 ± 3.49 | 6.35 ± 0.93 | .0484 ± .0088 | .0919 ± .0210 | 4.18 ± 0.80 | **8.36 ± 1.62** |
| **FairGRPO (FairMedGemma)** | **12.95 ± 1.82** | 6.84 ± 0.24 | 5.53 ± 0.29 | **.0379 ± .0005** | **.0724 ± .0004** | **4.11 ± 0.04** | 8.53 ± 0.11 |

| Training Method | Perf. Metrics | | Combined | |
|---|---|---|---|---|
| | Acc ↑ | F1 ↑ | Acc$_{ES}$ ↑ | F1$_{ES}$ ↑ |
| **Qwen-2.5-VL-7B** | | | | |
| Re++ (Hu, 2025) | 75.31 ± 1.82 | .2599 ± .0065 | 71.69 ± 1.39 | .2518 ± .0063 |
| RLOO (Ahmadian et al., 2024) | 78.22 ± 0.06 | .2523 ± .0013 | 74.59 ± 0.18 | .2443 ± .0014 |
| GRPO (Shao et al., 2024) | 78.40 ± 0.69 | .2601 ± .0131 | 76.21 ± 0.91 | .2425 ± .0017 |
| GRPO+RS (Puyol-Antón et al., 2021) | 75.61 ± 2.96 | .2580 ± .0021 | 71.67 ± 2.92 | .2500 ± .0018 |
| **FairGRPO** | **78.52 ± 0.31** | **.2657 ± .0036** | **77.14 ± 0.29** | **.2602 ± .0020** |
| **MedGemma-4B** | | | | |
| Re++ (Hu, 2025) | 78.76 ± 0.22 | .3105 ± .0179 | 75.41 ± 0.09 | .2966 ± .0191 |
| RLOO (Ahmadian et al., 2024) | 79.76 ± 0.16 | .3237 ± .0019 | 76.48 ± 0.20 | .3056 ± .0021 |
| GRPO (Shao et al., 2024) | 79.38 ± 0.15 | .3134 ± .0118 | 76.19 ± 0.12 | .3017 ± .0101 |
| GRPO+RS (Puyol-Antón et al., 2021) | 79.02 ± 0.15 | .2825 ± .0052 | 75.84 ± 0.30 | .2711 ± .0046 |
| GRPO+DRO (Sagawa et al., 2019) | 80.17 ± 0.31 | .3146 ± .0177 | 76.69 ± 0.48 | .3009 ± .0173 |
| **FairGRPO$_{ND}$** | 78.82 ± 0.58 | **.3484 ± .0041** | 75.67 ± 1.14 | **.3323 ± .0011** |
| **FairGRPO (FairMedGemma)** | **80.40 ± 0.03** | .3275 ± .0007 | **77.23 ± 0.01** | .3155 ± .0006 |

Table 19: **RQ1: Fairness and performance metrics for CheXpert dataset.** For fairness metrics, lower values are better and are indicated by ↓. For performance and combined metrics, higher values are better and are indicated by ↑. Bold values indicate the best result in each column.

| Training Method | Fairness Metrics | | | | | | | Perf. Metrics | | Combined | |
|---|---|---|---|---|---|---|---|---|---|---|---|
| | PP ↓ | EOD ↓ | FPR$_{Diff}$ ↓ | $\sigma_{F1}$ ↓ | $\Delta$F1 ↓ | $\sigma_{Acc}$ ↓ | $\Delta$Acc ↓ | Acc ↑ | F1 ↑ | Acc$_{ES}$ ↑ | F1$_{ES}$ ↑ |
| **Qwen-2.5-VL-7B** | | | | | | | | | | | |
| Re++ | 5.13$_{\pm 2.55}$ | 3.74$_{\pm 1.47}$ | 1.18$_{\pm 0.35}$ | .0148$_{\pm .0082}$ | .0282$_{\pm .0142}$ | 5.86$_{\pm 0.36}$ | 11.94$_{\pm 0.72}$ | 77.30$_{\pm 0.39}$ | .1149$_{\pm .0071}$ | 75.18$_{\pm 0.35}$ | .1257$_{\pm .0095}$ |
| RLOO | 20.32$_{\pm 7.47}$ | 6.25$_{\pm 4.87}$ | 4.09$_{\pm 1.37}$ | .0259$_{\pm .0164}$ | .0529$_{\pm .0388}$ | 5.42$_{\pm 0.07}$ | 10.90$_{\pm 0.07}$ | 77.74$_{\pm 0.73}$ | .1467$_{\pm .0100}$ | 75.67$_{\pm 0.57}$ | .1621$_{\pm .0215}$ |
| GRPO | 12.99$_{\pm 1.45}$ | 4.49$_{\pm 0.08}$ | 4.97$_{\pm 0.03}$ | .0194$_{\pm .0042}$ | .0386$_{\pm .0028}$ | 5.23$_{\pm 1.11}$ | 10.25$_{\pm 2.60}$ | 77.79$_{\pm 0.65}$ | .1443$_{\pm .0093}$ | 75.56$_{\pm 0.87}$ | .1572$_{\pm .0074}$ |
| GRPO+DRO | 11.31$_{\pm 2.86}$ | 3.91$_{\pm 1.31}$ | 1.80$_{\pm 0.09}$ | .0177$_{\pm .0034}$ | .0339$_{\pm .0051}$ | 6.38$_{\pm 0.31}$ | 13.45$_{\pm 0.78}$ | 76.91$_{\pm 0.41}$ | .1052$_{\pm .0042}$ | 74.95$_{\pm 0.50}$ | .1168$_{\pm .0061}$ |
| FairGRPO | 6.10$_{\pm 1.94}$ | 7.80$_{\pm 3.79}$ | 3.96$_{\pm 0.19}$ | .0234$_{\pm .0051}$ | .0439$_{\pm .0152}$ | 4.99$_{\pm 0.28}$ | 9.60$_{\pm 0.28}$ | 78.62$_{\pm 0.27}$ | .1372$_{\pm .0093}$ | 76.27$_{\pm 0.29}$ | .1510$_{\pm .0110}$ |
| **MedGemma-4B** | | | | | | | | | | | |
| Re++ | 23.93$_{\pm 12.50}$ | 9.09$_{\pm 1.74}$ | 4.61$_{\pm 1.52}$ | .0480$_{\pm .0020}$ | .0801$_{\pm .0051}$ | 3.86$_{\pm 2.46}$ | 8.37$_{\pm 4.93}$ | 78.53$_{\pm 2.18}$ | .2640$_{\pm .0083}$ | 77.10$_{\pm 2.66}$ | .2880$_{\pm .0069}$ |
| RLOO | 9.54$_{\pm 3.39}$ | 17.62$_{\pm 0.81}$ | 5.51$_{\pm 0.85}$ | .0817$_{\pm .0029}$ | .1465$_{\pm .0060}$ | 4.79$_{\pm 0.25}$ | 10.44$_{\pm 0.78}$ | 81.85$_{\pm 0.15}$ | .3354$_{\pm .0046}$ | 80.57$_{\pm 0.06}$ | .3827$_{\pm .0051}$ |
| GRPO | 5.61$_{\pm 3.84}$ | 8.32$_{\pm 0.99}$ | 3.12$_{\pm 0.51}$ | .0359$_{\pm .0071}$ | .0613$_{\pm .0147}$ | 5.22$_{\pm 0.82}$ | 11.18$_{\pm 1.67}$ | 82.22$_{\pm 0.53}$ | .2669$_{\pm .0038}$ | 80.73$_{\pm 0.55}$ | .2988$_{\pm .0081}$ |
| GRPO+RS | 13.48$_{\pm 2.94}$ | 12.39$_{\pm 2.10}$ | 3.76$_{\pm 0.49}$ | .0633$_{\pm .0175}$ | .1141$_{\pm .0283}$ | 4.43$_{\pm 0.15}$ | 9.81$_{\pm 0.01}$ | 83.64$_{\pm 0.31}$ | .3191$_{\pm .0019}$ | 82.55$_{\pm 0.21}$ | .3607$_{\pm .0022}$ |
| GRPO+DRO | 8.41$_{\pm 2.20}$ | 9.56$_{\pm 2.72}$ | 3.32$_{\pm 0.31}$ | .0587$_{\pm .0212}$ | .1030$_{\pm .0444}$ | 4.62$_{\pm 0.47}$ | 10.06$_{\pm 1.00}$ | 83.85$_{\pm 0.80}$ | .3230$_{\pm .0032}$ | 82.90$_{\pm 0.72}$ | .3664$_{\pm .0032}$ |
| FairGRPO$_{ND}$ | 20.42$_{\pm 3.13}$ | 7.88$_{\pm 1.17}$ | 6.89$_{\pm 0.37}$ | .0321$_{\pm .0008}$ | .0583$_{\pm .0055}$ | 4.70$_{\pm 0.35}$ | 9.89$_{\pm 0.98}$ | 81.30$_{\pm 1.02}$ | .3445$_{\pm .0093}$ | 79.94$_{\pm 1.00}$ | .3772$_{\pm .0022}$ |
| FairGRPO | 17.69$_{\pm 0.01}$ | 9.75$_{\pm 0.38}$ | 3.71$_{\pm 0.55}$ | .0501$_{\pm .0041}$ | .0836$_{\pm .0101}$ | 4.04$_{\pm 0.07}$ | 8.90$_{\pm 0.11}$ | 83.85$_{\pm 0.22}$ | .3220$_{\pm .0090}$ | 82.64$_{\pm 0.22}$ | .3605$_{\pm .0151}$ |

Table 20: **RQ1: Fairness and performance metrics for HAM10000 dataset.** For fairness metrics, lower values are better and are indicated by ↓. For performance and combined metrics, higher values are better and are indicated by ↑. Bold values indicate the best result in each column.

| Training Method | Fairness Metrics | | | | | | | Perf. Metrics | | Combined | |
|---|---|---|---|---|---|---|---|---|---|---|---|
| | PP↓ | EOD↓ | FPR_Diff↓ | $\sigma_{F1}$↓ | ΔF1↓ | $\sigma_{Acc}$↓ | ΔAcc↓ | Acc↑ | F1↑ | Acc_ES↑ | F1_ES↑ |
| **Qwen-2.5-VL-7B** | | | | | | | | | | | |
| Re++ | 22.90±0.28 | 8.56±3.49 | 4.72±1.63 | .0505±.0048 | .1135±.0009 | 5.97±1.49 | 12.28±2.31 | 84.59±2.76 | .2167±.0216 | 83.21±3.06 | .2178±.0150 |
| RLOO | 44.69±10.42 | 5.64±0.08 | 5.81±0.01 | .0469±.0020 | .1054±.0045 | 6.96±0.13 | 13.77±0.01 | 86.57±0.04 | .1974±.0000 | 85.43±0.07 | .2022±.0012 |
| GRPO | 35.65±1.72 | 7.25±0.13 | 7.08±0.09 | .0514±.0002 | .1149±.0013 | 6.87±0.06 | 13.75±0.12 | 86.55±0.10 | .1860±.0137 | 85.36±0.12 | .1862±.0094 |
| GRPO+DRO | 36.69±9.50 | 8.37±2.25 | 6.11±1.63 | .0471±.0041 | .0999±.0002 | 5.58±1.90 | 11.03±3.71 | 83.54±4.18 | .2140±.0253 | 82.49±4.08 | .2159±.0256 |
| FairGRPO | 26.56±8.25 | 8.21±1.08 | 5.98±3.16 | .0325±.0132 | .0639±.0216 | 5.92±0.42 | 11.64±1.12 | 86.67±0.64 | .2738±.0006 | 85.71±0.44 | .2581±.0061 |
| **MedGemma-4B** | | | | | | | | | | | |
| Re++ | 24.42±5.85 | 6.33±0.39 | 6.22±1.82 | .0417±.0037 | .0958±.0094 | 4.66±2.10 | 9.46±3.98 | 85.50±1.66 | .2172±.0138 | 84.55±1.52 | .2220±.0194 |
| RLOO | 30.65±4.83 | 6.08±1.59 | 6.79±0.96 | .0318±.0019 | .0708±.0008 | 6.51±0.04 | 12.81±0.09 | 86.72±0.11 | .2319±.0065 | 85.68±0.10 | .2288±.0094 |
| GRPO | 33.41±11.32 | 6.10±0.71 | 6.97±2.05 | .0455±.0073 | .0992±.0151 | 6.49±0.03 | 12.93±0.41 | 86.86±0.02 | .2492±.0020 | 85.73±0.14 | .2491±.0111 |
| GRPO+RS | 29.83±6.08 | 6.19±0.89 | 5.24±2.06 | .0501±.0083 | .1111±.0223 | 6.81±0.55 | 13.65±0.91 | 86.07±0.57 | .1856±.0373 | 85.09±0.25 | .1898±.0313 |
| GRPO+DRO | 22.42±5.99 | 8.89±5.37 | 5.98±1.88 | .0655±.0326 | .1455±.0727 | 6.72±0.09 | 13.36±0.43 | 86.45±0.19 | .2285±.0080 | 85.46±0.10 | .2394±.0215 |
| FairGRPO_ND | 13.95±2.36 | 7.23±2.80 | 5.57±0.94 | .0383±.0174 | .0702±.0225 | 5.05±0.36 | 9.96±0.65 | 85.55±0.27 | .2743±.0136 | 84.60±0.38 | .2662±.0027 |
| FairGRPO | 23.80±13.18 | 6.03±0.97 | 5.76±1.99 | .0445±.0038 | .0978±.0036 | 6.86±0.04 | 13.55±0.14 | 86.68±0.03 | .2170±.0096 | 85.56±0.01 | .2166±.0134 |

Table 21: **RQ1: Fairness and performance metrics for ISIC2020 dataset.** For fairness metrics, lower values are better and are indicated by ↓. For performance and combined metrics, higher values are better and are indicated by ↑. Bold values indicate the best result in each column.

| Training Method | Fairness Metrics | | | | | | | Perf. Metrics | | Combined | |
|---|---|---|---|---|---|---|---|---|---|---|---|
| | PP↓ | EOD↓ | FPR_Diff↓ | $\sigma_{F1}$↓ | ΔF1↓ | $\sigma_{Acc}$↓ | ΔAcc↓ | Acc↑ | F1↑ | Acc_ES↑ | F1_ES↑ |
| **Qwen-2.5-VL-7B** | | | | | | | | | | | |
| Re++ | 21.51±23.07 | 3.10±3.06 | 3.06±3.00 | .0196±.0068 | .0412±.0165 | 3.04±0.84 | 6.56±2.03 | 96.70±2.16 | .5205±.0212 | 95.51±2.30 | .5183±.0267 |
| RLOO | 2.59±0.02 | 0.58±0.47 | 0.47±0.67 | .0110±.0054 | .0222±.0103 | 2.51±0.08 | 5.26±0.17 | 98.20±0.03 | .5004±.0073 | 97.08±0.07 | .4957±.0052 |
| GRPO | 26.52±33.86 | 0.01±0.02 | 0.01±0.02 | .0066±.0000 | .0137±.0000 | 2.47±0.01 | 5.17±0.01 | 98.23±0.00 | .4955±.0000 | 97.14±0.00 | .4926±.0000 |
| GRPO+DRO | 14.75±15.01 | 5.84±7.91 | 5.83±7.92 | .0155±.0126 | .0340±.0288 | 3.29±1.13 | 7.00±2.67 | 94.78±4.26 | .5056±.0159 | 93.69±4.42 | .5053±.0192 |
| FairGRPO | 2.59±0.02 | 0.12±0.17 | 0.00±0.00 | .0068±.0003 | .0141±.0006 | 2.50±0.05 | 5.23±0.10 | 98.14±0.14 | .4950±.0007 | 97.04±0.15 | .4921±.0008 |
| **MedGemma-4B** | | | | | | | | | | | |
| Re++ | 2.58±0.01 | 0.49±0.64 | 0.47±0.67 | .0107±.0060 | .0216±.0114 | 2.46±0.04 | 5.14±0.01 | 98.24±0.03 | .5006±.0072 | 97.15±0.02 | .4961±.0050 |
| RLOO | 2.58±0.00 | 0.14±0.05 | 0.00±0.00 | .0063±.0002 | .0132±.0001 | 2.41±0.04 | 5.05±0.01 | 98.10±0.01 | .4948±.0000 | 97.02±0.02 | .4920±.0001 |
| GRPO | 16.79±20.39 | 1.53±1.82 | 2.80±0.07 | .0181±.0177 | .0369±.0358 | 2.38±0.17 | 4.97±0.36 | 97.95±0.08 | .5134±.0278 | 96.90±0.02 | .5050±.0192 |
| GRPO+RS | 2.58±0.00 | 0.00±0.00 | 0.00±0.00 | .0066±.0000 | .0137±.0000 | 2.47±0.00 | 5.16±0.00 | 98.24±0.00 | .4956±.0000 | 97.14±0.00 | .4926±.0000 |
| GRPO+DRO | 2.58±0.00 | 0.02±0.03 | 0.00±0.00 | .0065±.0001 | .0136±.0001 | 2.46±0.01 | 5.15±0.00 | 98.23±0.01 | .4955±.0001 | 97.14±0.01 | .4926±.0000 |
| FairGRPO_ND | 30.75±5.81 | 3.38±0.61 | 3.26±0.74 | .0291±.0013 | .0570±.0051 | 2.31±0.13 | 4.91±0.31 | 97.89±0.07 | .5433±.0172 | 96.81±0.03 | .5322±.0215 |
| FairGRPO | 2.58±0.00 | 0.00±0.00 | 0.00±0.00 | .0066±.0000 | .0137±.0000 | 2.47±0.00 | 5.16±0.00 | 98.24±0.00 | .4956±.0000 | 97.14±0.00 | .4926±.0000 |

Table 22: **RQ1: Fairness and performance metrics for PAD-UFES-20 dataset.** For fairness metrics, lower values are better and are indicated by ↓. For performance and combined metrics, higher values are better and are indicated by ↑. Bold values indicate the best result in each column.

| Training Method | Fairness Metrics | | | | | | | Perf. Metrics | | Combined | |
|---|---|---|---|---|---|---|---|---|---|---|---|
| | PP↓ | EOD↓ | FPR_Diff↓ | $\sigma_{F1}$↓ | ΔF1↓ | $\sigma_{Acc}$↓ | ΔAcc↓ | Acc↑ | F1↑ | Acc_ES↑ | F1_ES↑ |
| **Qwen-2.5-VL-7B** | | | | | | | | | | | |
| Re++ | 34.76±1.25 | 13.01±5.64 | 10.14±0.71 | .0589±.0255 | .1231±.0543 | 3.73±0.78 | 7.58±1.99 | 77.96±0.07 | .3129±.0035 | 79.42±0.20 | .3121±.0122 |
| RLOO | 36.69±0.65 | 16.74±1.42 | 10.12±0.27 | .0851±.0019 | .1788±.0034 | 6.15±0.27 | 12.13±0.09 | 77.12±0.37 | .2672±.0078 | 79.97±0.31 | .2788±.0081 |
| GRPO | 37.46±6.33 | 16.50±4.29 | 13.89±6.40 | .0826±.0120 | .1686±.0258 | 4.90±1.22 | 10.18±2.89 | 76.70±0.67 | .2614±.0309 | 78.72±0.96 | .2684±.0241 |
| GRPO+DRO | 37.38±2.46 | 15.56±2.88 | 9.36±1.06 | .0680±.0209 | .1346±.0330 | 4.20±0.03 | 8.74±0.22 | 78.22±0.30 | .3271±.0088 | 79.94±0.69 | .3229±.0142 |
| FairGRPO | 24.50±9.47 | 8.26±3.84 | 10.14±1.15 | .0516±.0031 | .1075±.0115 | 1.92±0.30 | 4.09±0.46 | 80.31±2.52 | .2995±.0547 | 78.70±2.74 | .2923±.0486 |
| **MedGemma-4B** | | | | | | | | | | | |
| Re++ | 39.59±6.57 | 15.75±2.68 | 10.58±0.65 | .1040±.0202 | .1891±.0306 | 4.38±0.41 | 8.26±0.77 | 79.16±1.55 | .3089±.0392 | 80.53±1.21 | .3153±.0466 |
| RLOO | 38.93±8.56 | 9.86±0.49 | 9.92±0.11 | .0778±.0041 | .1538±.0086 | 3.34±0.27 | 6.88±0.12 | 77.43±0.52 | .2688±.0088 | 79.45±0.74 | .2794±.0121 |
| GRPO | 36.94±0.17 | 9.57±0.24 | 9.50±0.66 | .0774±.0006 | .1580±.0078 | 3.28±0.28 | 6.80±0.53 | 76.78±0.48 | .2427±.0229 | 78.99±0.28 | .2578±.0206 |
| GRPO+RS | 29.68±2.63 | 13.93±3.97 | 16.60±10.32 | .0733±.0061 | .1478±.0011 | 3.13±0.81 | 6.49±2.93 | 76.91±3.92 | .2530±.0316 | 76.24±1.97 | .2498±.0148 |
| GRPO+DRO | 38.00±10.53 | 11.62±0.03 | 13.91±5.33 | .0713±.0061 | .1481±.0176 | 3.65±0.92 | 7.60±2.26 | 77.46±0.26 | .2408±.0024 | 79.34±0.96 | .2394±.0077 |
| FairGRPO_ND | 45.17±2.88 | 12.95±2.69 | 8.81±0.93 | .0808±.0062 | .1643±.0185 | 3.89±1.39 | 8.12±2.78 | 78.77±0.70 | .2950±.0340 | 80.90±1.12 | .2996±.0281 |
| FairGRPO | 18.00±0.77 | 11.41±1.04 | 8.61±0.16 | .0734±.0040 | .1349±.0070 | 2.44±0.06 | 5.02±0.02 | 83.51±0.00 | .3620±.0013 | 82.74±0.09 | .3448±.0044 |

Table 23: **RQ1: Fairness and performance metrics for Hemorrhage dataset.** For fairness metrics, lower values are better and are indicated by ↓. For performance and combined metrics, higher values are better and are indicated by ↑. Bold values indicate the best result in each column.

| Training Method | Fairness Metrics | | | | | | | Perf. Metrics | | Combined | |
|---|---|---|---|---|---|---|---|---|---|---|---|
| | PP↓ | EOD↓ | FPR_Diff↓ | $\sigma_{F1}$↓ | ΔF1↓ | $\sigma_{Acc}$↓ | ΔAcc↓ | Acc↑ | F1↑ | Acc_ES↑ | F1_ES↑ |
| **Qwen-2.5-VL-7B** | | | | | | | | | | | |
| Re++ | 14.94±0.12 | 9.35±5.58 | 9.35±5.58 | .0557±.0345 | .0890±.0574 | 4.20±0.42 | 7.29±0.48 | 75.83±3.71 | .4822±.0246 | 75.28±2.94 | .4926±.0332 |
| RLOO | 13.23±10.27 | 1.74±0.41 | 1.74±0.41 | .0213±.0049 | .0397±.0079 | 6.51±1.39 | 12.22±2.25 | 81.26±0.41 | .4483±.0013 | 79.53±0.48 | .4420±.0018 |
| GRPO | 12.58±7.06 | 3.74±1.44 | 3.74±1.44 | .0302±.0269 | .0495±.0455 | 5.48±0.97 | 9.90±1.27 | 80.78±2.75 | .4805±.0314 | 79.28±2.79 | .4811±.0349 |
| GRPO+DRO | 16.38±5.61 | 5.96±1.93 | 5.96±1.93 | .0407±.0056 | .0639±.0146 | 6.63±1.20 | 12.12±1.93 | 79.71±2.06 | .4886±.0167 | 78.52±1.91 | .4923±.0160 |
| FairGRPO | 34.02±21.84 | 1.59±2.00 | 1.59±2.00 | .0263±.0096 | .0501±.0212 | 6.14±0.09 | 10.91±0.08 | 83.30±0.00 | .5011±.0660 | 81.84±0.30 | .4970±.0685 |
| **MedGemma-4B** | | | | | | | | | | | |
| Re++ | 21.38±3.64 | 6.35±1.18 | 6.00±0.94 | .0622±.0173 | .1067±.0456 | 5.08±0.20 | 8.27±0.85 | 83.69±0.27 | .6382±.0462 | 83.92±0.14 | .6423±.0625 |
| RLOO | 20.79±1.64 | 8.61±1.25 | 8.30±1.69 | .0643±.0013 | .1166±.0152 | 6.17±0.03 | 10.80±0.32 | 84.61±0.62 | .6513±.0026 | 84.12±0.11 | .6532±.0136 |
| GRPO | 17.42±5.31 | 9.66±3.24 | 10.22±0.71 | .0690±.0182 | .1172±.0220 | 5.18±0.36 | 8.65±1.00 | 83.74±0.48 | .6370±.0317 | 84.08±0.21 | .6508±.0462 |
| GRPO+RS | 36.49±1.33 | 11.83±0.29 | 4.88±0.04 | .0661±.0041 | .1183±.0000 | 6.14±0.05 | 11.32±0.14 | 82.62±0.41 | .5022±.0023 | 81.04±0.33 | .4808±.0017 |
| GRPO+DRO | 19.21±1.70 | 7.31±1.03 | 7.31±1.03 | .0721±.0058 | .1132±.0126 | 4.33±0.59 | 6.45±1.08 | 84.56±0.96 | .7238±.0071 | 85.62±0.62 | .7460±.0066 |
| FairGRPO_ND | 17.15±0.27 | 10.06±0.64 | 9.20±0.26 | .0837±.0089 | .1378±.0211 | 4.71±0.33 | 7.09±0.44 | 81.41±1.58 | .6957±.0027 | 82.87±1.13 | .7208±.0035 |
| FairGRPO | 18.01±0.12 | 7.33±0.68 | 7.33±0.68 | .0539±.0078 | .0909±.0103 | 3.54±0.13 | 5.88±0.53 | 84.08±0.27 | .6951±.0008 | 84.61±0.08 | .7083±.0025 |

Table 24: **RQ1: Fairness and performance metrics for VinDr dataset.** For fairness metrics, lower values are better and are indicated by ↓. For performance and combined metrics, higher values are better and are indicated by ↑. Bold values indicate the best result in each column.

| Training Method | Fairness Metrics | | | | | | | Perf. Metrics | | Combined | |
|---|---|---|---|---|---|---|---|---|---|---|---|
| | PP ↓ | EOD ↓ | FPR$_{Diff}$ ↓ | $\sigma_{F1}$ ↓ | $\Delta$F1 ↓ | $\sigma_{Acc}$ ↓ | $\triangle$Acc ↓ | Acc ↑ | F1 ↑ | Acc$_{ES}$ ↑ | F1$_{ES}$ ↑ |
| **Qwen-2.5-VL-7B** | | | | | | | | | | | |
| **Re++** | $17.41_{\pm4.38}$ | $8.87_{\pm6.14}$ | $16.16_{\pm8.70}$ | $.0261_{\pm.0030}$ | $.0581_{\pm.0016}$ | $12.58_{\pm2.80}$ | $26.64_{\pm5.98}$ | $75.51_{\pm6.11}$ | $.1676_{\pm.0447}$ | $68.59_{\pm3.89}$ | $.1530_{\pm.0321}$ |
| **RLOO** | $29.55_{\pm0.84}$ | $10.91_{\pm3.42}$ | $15.38_{\pm2.59}$ | $.0314_{\pm.0072}$ | $.0742_{\pm.0162}$ | $10.68_{\pm1.82}$ | $23.76_{\pm3.75}$ | $86.62_{\pm0.16}$ | $.1704_{\pm.0051}$ | $79.25_{\pm1.16}$ | $.1705_{\pm.0026}$ |
| **GRPO** | $31.91_{\pm2.14}$ | $13.33_{\pm0.00}$ | $12.50_{\pm1.63}$ | $.0389_{\pm.0040}$ | $.0918_{\pm.0087}$ | $9.33_{\pm0.09}$ | $20.89_{\pm0.19}$ | $86.58_{\pm0.21}$ | $.1689_{\pm.0030}$ | $80.01_{\pm0.10}$ | $.1745_{\pm.0030}$ |
| **GRPO+DRO** | $20.81_{\pm15.05}$ | $8.29_{\pm2.30}$ | $16.00_{\pm0.52}$ | $.0345_{\pm.0087}$ | $.0730_{\pm.0189}$ | $12.43_{\pm2.80}$ | $26.47_{\pm4.88}$ | $76.77_{\pm8.94}$ | $.1625_{\pm.0524}$ | $69.69_{\pm9.11}$ | $.1527_{\pm.0542}$ |
| **FairGRPO** | $14.19_{\pm5.07}$ | $13.35_{\pm0.00}$ | $13.36_{\pm0.04}$ | $.0369_{\pm.0002}$ | $.0856_{\pm.0000}$ | $9.44_{\pm0.01}$ | $21.20_{\pm0.01}$ | $86.82_{\pm0.02}$ | $.1608_{\pm.0004}$ | $80.12_{\pm0.02}$ | $.1696_{\pm.0003}$ |
| **MedGemma-4B** | | | | | | | | | | | |
| **Re++** | $30.24_{\pm16.40}$ | $16.44_{\pm11.83}$ | $11.94_{\pm0.11}$ | $.0620_{\pm.0433}$ | $.1353_{\pm.1031}$ | $10.61_{\pm0.60}$ | $23.46_{\pm0.86}$ | $85.02_{\pm0.87}$ | $.1918_{\pm.0129}$ | $78.05_{\pm0.92}$ | $.2096_{\pm.0052}$ |
| **RLOO** | $40.63_{\pm14.04}$ | $30.16_{\pm0.22}$ | $8.03_{\pm0.06}$ | $.1524_{\pm.0001}$ | $.3201_{\pm.0012}$ | $6.79_{\pm0.08}$ | $15.58_{\pm0.06}$ | $86.73_{\pm0.15}$ | $.1645_{\pm.0023}$ | $81.44_{\pm0.10}$ | $.2297_{\pm.0003}$ |
| **GRPO** | $36.08_{\pm2.85}$ | $8.90_{\pm2.78}$ | $4.23_{\pm2.71}$ | $.0250_{\pm.0044}$ | $.0542_{\pm.0088}$ | $6.77_{\pm0.56}$ | $15.42_{\pm1.09}$ | $83.80_{\pm1.26}$ | $.2190_{\pm.0065}$ | $79.26_{\pm0.81}$ | $.2323_{\pm.0065}$ |
| **GRPO+RS** | $59.80_{\pm14.91}$ | $11.43_{\pm0.35}$ | $2.97_{\pm0.01}$ | $.0357_{\pm.0017}$ | $.0784_{\pm.0001}$ | $6.44_{\pm0.03}$ | $14.93_{\pm0.06}$ | $86.06_{\pm0.08}$ | $.1850_{\pm.0055}$ | $81.10_{\pm0.11}$ | $.2041_{\pm.0056}$ |
| **GRPO+DRO** | $11.43_{\pm1.59}$ | $11.37_{\pm0.25}$ | $5.84_{\pm3.81}$ | $.0332_{\pm.0056}$ | $.0725_{\pm.0137}$ | $7.29_{\pm0.95}$ | $16.72_{\pm2.03}$ | $86.48_{\pm0.06}$ | $.1863_{\pm.0024}$ | $80.98_{\pm0.38}$ | $.1940_{\pm.0024}$ |
| **FairGRPO**$_{ND}$ | $46.67_{\pm12.41}$ | $22.13_{\pm26.94}$ | $10.69_{\pm7.72}$ | $.0751_{\pm.0815}$ | $.1555_{\pm.1614}$ | $8.60_{\pm6.11}$ | $18.55_{\pm12.38}$ | $81.84_{\pm0.44}$ | $.2128_{\pm.0262}$ | $76.65_{\pm2.52}$ | $.2428_{\pm.0262}$ |
| **FairGRPO** | $10.61_{\pm0.00}$ | $13.33_{\pm0.00}$ | $13.33_{\pm0.00}$ | $.0370_{\pm.0000}$ | $.0856_{\pm.0000}$ | $9.44_{\pm0.00}$ | $21.21_{\pm0.00}$ | $86.82_{\pm0.00}$ | $.1606_{\pm.0000}$ | $80.12_{\pm0.00}$ | $.1694_{\pm.0000}$ |

