# OpenReview forum: "FairGRPO: Fair Reinforcement Learning for Equitable Clinical Reasoning"
_ICLR.cc/2026/Conference — Submitted to ICLR 2026_

### Official Review · Reviewer_PPRM · 2025-10-31

**Soundness:** 3
**Presentation:** 2
**Contribution:** 2
**Rating:** 4
**Confidence:** 4

**Summary:**

In this manuscript, the authors focus on an important but under-explored task, fairness in medical artificial intelligence systems. The authors aim to improve the fairness and effectiveness during the reinforcement learning phase of vision language models. And they propose to modify the advantages to address the limitations. Through experiments on 7 clinical diagnostic datasets with 2 representative VLMs, the authors demonstrate the effectiveness of their proposed method.

**Strengths:**

### **Strengths**
1. This manuscript focuses on a under-explored but important task, mitigating unfairness in the medical reasoning.

2. The experimental datasets are comprehensive. The authors collect 7 clinical diagnostic dataset spanning 5 clinical modalities, which is can adequately verify whether the method is effective.

3. The experimental section considers a comprehensive set of dimensions, including effectiveness, fairness, efficiency, case study, and others.

**Weaknesses:**

### **Weaknesses**

1. My primary concern regarding this paper is the lack of sufficient novelty in the proposed approach. The modification of advantages is easy and lack of theoretical analysis. And the clustering is a common approach in fairness machine learning without demographic information.

2. The authors conduct experiments on Qwen-2.5-VL-7B and MedGemma-4B. Although these two models are representative, it is better to conduct more experiments on more VLMs.

**Questions:**

### **Questions**

1. Could the authors provide more theoretical or empirical analysis of the proposed method/motivation?

2. How about the performance of the proposed FairGRPO on the other mainstream VLMs?

3. Could the authors provide more case studies to prove the importance of fairness in medical AI systems?

---

> ### Author Response · Authors · 2025-11-21
> **Response to reviewer PPRM**
>
> We thank reviewer PPRM for the constructive feedback and patience as we gather data from more experiments addressing the remaining points.
>
> > PPRM.C1: My primary concern regarding this paper is the lack of sufficient novelty in the proposed approach.
>
> Our work constitutes the first attempt to improve fairness in medical foundation models with the use of a semi-supervised reinforcement-learning-based mechanism. A notable advantage of FairGRPO is that it remains effective in both labeled and unlabeled demographic settings. The novelty of our approach lies in leveraging the reward-based features instead of the extracted features as the feature representation used when performing clustering, thus providing an intuitive manner to cluster samples based on their task-specific difficulty patterns. The proposed adaptive importance weighting based on demographic representation and task difficulty methodology constitutes one the first fair critic-free reinforcement learning-based method that consistently improves overall performance and fairness.
>
> We argue that the method’s simplicity is a core strength, producing a neat, intuitive, and effective solution. Its effectiveness is further demonstrated by the non-trivial improvements on the performance–Pareto frontier (Figure 2e) and by the performance and fairness results in Table 2. Beyond the main algorithmic contribution outlined above, we further publicly release the fairness-aware vision clinical model, FairMedGemma-4B, to facilitate further experimentation.
>
> > PPRM.C2: The modification of advantages is easy and lack of theoretical analysis.
>
> We appreciate the reviewer's question about theoretical guarantees. We clarify that FairGRPO inherits the convergence properties of GRPO while adding demographic-aware scaling. The key insight is that our scaling mechanism preserves the essential structure needed for convergence.
>
> FairGRPO maintains GRPO's advantage normalization structure with $\hat{A}\_{(q,i,t)}^{\text{FairGRPO}} = \frac{s\_{(q,i,t)}^{\text{scaled}}}{\sigma\_{\text{batch}}}$, where the batch-wise renormalization ensures zero mean and unit variance, which is a sufficient condition for GRPO's convergence properties [1]. The demographic scaling factors $T\_{(g,t)}$ and $T\_{(\gamma,g,t)}$ are bounded by design through the square root of sample counts and mean rewards, preventing unbounded scaling that could destabilize training. This bounded scaling ensures that the gradient magnitudes remain within reasonable ranges throughout optimization, preserving the stability properties of the original GRPO algorithm.
>
> In addition, we designed our inverse temperature scaling to prevent overcompensation through its mathematical structure. For a group $\gamma$ with representation $N\_{(\gamma,g,t)}$ and performance $\bar{r}\_{(\gamma,g,t)}$, the scaling factor $T\_{(\gamma,g,t)} = \sqrt{N\_{(\gamma,g,t)} \cdot \bar{r}\_{(\gamma,g,t)}}$ creates a self-regulating mechanism. As minority groups improve (increasing $\bar{r}\_{(\gamma,g,t)}$), their scaling factor increases, reducing the amplification. This creates a natural equilibrium where groups converge toward similar performance levels without overcompensation.
>
> Lastly, while formal convergence proofs for policy gradient methods with adaptive weighting remain an open research question, our extensive experiments across 7 datasets provide strong empirical evidence. Figure 2(a-b) shows that FairGRPO maintains similar convergence trajectories to GRPO for overall metrics while simultaneously improving fairness metrics. The Pareto frontier analysis in Figure 2(e) further demonstrates that FairGRPO finds solutions that dominate GRPO in the performance-fairness space, suggesting the method converges to better optima rather than merely different ones. We acknowledge that providing formal convergence guarantees for adaptive importance weighting in policy optimization is an important direction for future theoretical work, similar to ongoing efforts in the broader fair RL literature.
>
> [1] Zhihong Shao, Peiyi Wang, Qihao Zhu, Runxin Xu, Junxiao Song, Xiao Bi, Haowei Zhang, Mingchuan Zhang, YK Li, Y Wu, et al. "Deepseekmath: Pushing the limits of mathematical reasoning in open language models." https://arxiv.org/abs/2402.03300

---

> ### Author Response · Authors · 2025-11-21
> **Response to reviewer PPRM - Continued 1**
>
> > PPRM.C3: And the clustering is a common approach in fairness machine learning without demographic information.
>
> The main contributions of our paper is (i) the adaptive importance weighting based on demographic representation and task difficulty as well as the (ii) public release of a fairness-aware vision clinical model, FairMedGemma-4B. Rather than proposing clustering as a contribution, we mainly adopted clustering to serve as a solution when demographic information is not available, given its wide acceptance within the community. The novelty of our approach lies in leveraging the reward-based features instead of the extracted features as the feature representation used when performing clustering, thus providing an intuitive manner to cluster samples based on their task-specific difficulty patterns.
>
> > PPRM.C5: Could the authors provide more theoretical or empirical analysis of the proposed method/motivation?
>
> We included a theoretical analysis of the method in A2 above. In addition, we conducted empirical analysis regarding the quality of learned clusters. When the demographic information is not passed to the model, the discovered groups have an average Adjusted Rand Index (ARI) of 0.109 and a Normalized Mutual Information (NMI) of 0.355, indicating positive correlation between the discovered groups and the ground truth demographic groups.
>
> On the scaling side, we see the model with discovered groups upscale samples from minority groups:
>
> | Demo Group | M,A1 | F,A2 | A2,F | M,A2 | M,A4 |
> |------|------|------|------|------|------|
> | Avg Scale | 0.031±0.037 | 0.078±0.160 | 0.083±0.124 | 0.089±0.170 | 0.110±0.189 |
> | A2,M | F,A4 | F,A3 | F,A1 | M,A3 | A2 |
> | 0.117±0.315 | 0.147±0.219 | 0.153±0.217 | 0.172±0.207 | 0.174±0.239 | 0.641±0.833 |
> | A3 | A3,M | A4 | A1 | A3,F | A1,F |
> | 0.863±1.184 | 1.046±1.110 | 1.414±2.071 | 1.515±1.533 | 2.955±8.435 | 3.024±0.000 |
>
> In the above table, the demographic group is defined as in the paper. For age, we
> create four groups using 25-year bins: a1 for ages 18-25, a2 for ages 26-50, a3 for ages 51-75, and a4 for ages 76 and above. M indicates males, and F denotes females.
>
> We see that even without ground truth demographic groups, the model tends to upscale minority samples, particularly female samples (A1, F scale of 3.024, A3, F scale of 2.955) and people of minority age groups (A1 scale = 1.515, A4 scale = 1.414). This validates our hypothesis that minority samples tend to be harder samples for the model throughout the training process, and upscaling these samples has a positive effect on the fairness of the model.
>
> > PPRM.C7: Could the authors provide more case studies to prove the importance of fairness in medical AI systems?
>
> We thank the reviewer for highlighting this crucial point. Our fairness metrics are specifically chosen to reflect direct clinical consequences. For instance, the 6.66% disparity in Equal Opportunity Difference from GRPO+RS means that for every 100 positive cases detected in majority groups, minority groups would have approximately 7 fewer cases correctly identified. In our dermoscopy datasets, this translates to missing 7 melanomas per 100 cases in elderly populations. This is a critical oversight given that melanoma's 5-year survival rate drops from 99% when caught early to 35% in advanced stages[1]. Similarly, the 23.76% Predictive Parity gap means that nearly 1 in 4 positive predictions for minority groups are false positives compared to majority groups. In mammography screening, this would subject minority women to approximately 24 additional unnecessary biopsies per 100 positive predictions, each carrying risks of complications, psychological distress, and financial burden.
> The qualitative cases in Figure 4 exemplify these failures. In the dermoscopy case, the baseline model fabricates a "central dot" feature, misdiagnosing Basal Cell Carcinoma as AKIEC in an 84-year-old female. This is a demographic where vanilla GRPO shows 28.91% lower diagnostic accuracy. On the other hand, the vanilla model downgrades diagnosis BIRAD 2 (benign, needs monitoring) to BIRAD 1 (completely normal) removes follow-up protocols for elderly women who already face 2-3x higher breast cancer risk. FairGRPO reduces these disparities by 27.2% in predictive parity and improves F1 scores by 12.49%, translating to approximately 13 more correct diagnoses per 100 cases in underserved populations. When employed clinically, the improved model has the potential to significantly improve the clinical outcome of minority populations and reduce the financial burden overall, thanks to the reduced misdiagnosis.
>
> [1] American Cancer Society. (2024). [Melanoma Skin Cancer Survival Rates](https://www.cancer.org/cancer/types/melanoma-skin-cancer/detection-diagnosis-staging/survival-rates-for-melanoma-skin-cancer-by-stage.html). Cancer Facts & Statistics.

---

> > ### Author Response · Authors · 2025-11-26
> > **Response to Reviewer PPRM - Continued 2**
> >
> > > PPRM.Q2: How about the performance of the proposed FairGRPO on the other mainstream VLMs? W2: The authors conduct experiments on Qwen-2.5-VL-7B and MedGemma-4B. Although these two models are representative, it is better to conduct more experiments on more VLMs.
> >
> > We agree that it is useful to validate the effectiveness of our method on more models, especially on larger models to prove its scalability. We expanded the experiment to Qwen2.5-VL-32B, comparing the performance and fairness of FairGRPO against GRPO+RS and GRPO+DRO with 3 separate training runs. The results are as follows:
> >
> > | Training Method | PP ↓ | EOD ↓ | FPR_Diff ↓ | σ_F1 ↓ | ΔF1 ↓ | σ_Acc ↓ | ΔAcc ↓ | Acc ↑ | F1 ↑ | Acc_ES ↑ | F1_ES ↑ |
> > |-----------------|------|-------|------------|--------|-------|---------|--------|-------|------|----------|---------|
> > | GRPO+RS | 24.65 | 7.889 | 7.561 | .0401 | .0786 | 4.470 | 8.999 | 77.73 | .2641 | 74.40 | .2539 |
> > | GRPO+DRO | **18.46** | 6.903 | 6.674 | **.0325** | .0631 | 4.043 | 8.156 | 77.67 | .2694 | 74.65 | .2609 |
> > | **FairGRPO** | 22.26 | **6.555** | **6.462** | .0343 | **.0622** | **3.855** | **7.834** | **77.99** | **.2792** | **75.10** | **.2699** |
> >
> > In short, we see that our method still performs optimally when the model scales to 32B, achieving optimal fairness as measured by EOD, FPR_Diff, ΔF1, σ_Acc, and ΔAcc; best performance as measured by Acc, and F1; and best performance-fairness trade-off as measured by Acc_ES, and F1_ES.
> >
> > We thank reviewer PPRM again for their patience as we gathered the results from this last large-scale experiment. If you have any further questions, please let us know.

---

### Official Review · Reviewer_TS5D · 2025-10-31

**Soundness:** 3
**Presentation:** 3
**Contribution:** 3
**Rating:** 6
**Confidence:** 3

**Summary:**

Modern medical AI systems may show performance disparities wrt demographic groups (e.g., by race, gender, age) because training data is heavily skewed toward majority groups. This paper addresses fairness in reinforcement learning for clinical reasoning. The paper proposes a method that weighs samples according to the representation of the demographic group (underrepresented groups get higher weight),  task difficulty, and data source. Whenever demographic labels are not available, clustering is used to create implicit demographic groups. The technique is used to train a multi-modal clinical model, FairMedGemma-4B and its performance is analyzed for different clinical tasks. The performance across demographic groups tends to be more homogeneous. In addition, the fairness continually improves during training.

**Strengths:**

One of the first works to embed group fairness considerations into the RL for clinical reasoning models

**Weaknesses:**

Not clear how the group fairness influences individual performance, which is what is most important.

**Questions:**

Do you have an intuition on how the group fairness policies affect individual metrics of fairness?

---

> ### Author Response · Authors · 2025-11-21
> **Response to reviewer TS5D**
>
> We sincerely thank reviewer TS5D for the overall positive feedback. Individual metrics of fairness typically dictate that similar individuals are treated similarly by the algorithm or model. Within the context of our algorithm, given that we are attempting to improve group fairness by amplifying learning signals from the (individually-similar) underrepresented or challenging groups, our proposed method naturally improves individual metrics of fairness whilst concurrently improving group fairness. This is evidenced by our results in Table 2. Moreover, an improvement in accuracy indisputably means that more individual samples are correctly classified, which leads to better individual-based outcomes.

---

> > ### Comment · Reviewer_TS5D · 2025-11-25
> >
> > Thank you for the clarification. I will maintain my scores.

---

### Official Review · Reviewer_Tq3Z · 2025-11-01

**Soundness:** 2
**Presentation:** 2
**Contribution:** 3
**Rating:** 4
**Confidence:** 4

**Summary:**

This paper proposes FairGRPO, a reinforcement learning algorithm that mitigates demographic bias in clinical vision–language models. FairGRPO adaptively re-weights learning signals based on representation, task difficulty and data source, and uses unsupervised clustering to infer latent demographic groups when labels are missing. Evaluated on seven multimodal clinical datasets with Qwen-2.5-VL-7B and MedGemma-4B, the method improves both fairness and accuracy over RL and fairness mitigation baselines.

**Strengths:**

1. This paper addresses fairness in reinforcement learning for multimodal clinical models, a critical but underexplored area. By focusing on fairness in critic-free RL (e.g., GRPO-style optimization), the work bridges the gap of fair ML for healthcare communities.
2. The proposed method of scaling advantages inversely by group representation and task difficulty is simple to compute and adds negligible runtime overhead.
3. Experiments span seven clinical datasets covering diverse imaging types, which demonstrates the generalization ability of FairGRPO.
4. The paper presents fairness trajectories during optimization, showing that FairGRPO progressively improves fairness instead of degrading it, as seen in baseline RL methods. The qualitative case studies vividly demonstrate how fairness-aware training improves the model’s reasoning trace and diagnostic correctness.

**Weaknesses:**

1. The theoretical grounding for the proposed fairness optimization is limited. It is unclear whether the scaling guarantees convergence or prevents overcompensation.
2. The paper employs K-means clustering on reward vectors to infer latent demographic groups, but does not analyze the robustness of this clustering step. The number of clusters, initialization, and metric choice could influence group assignments, potentially introducing instability or even amplifying unintended biases in underrepresented groups.
3. The design of group discovery lacks interpretability. The relation between the learned task-specific difficulty patterns and demographic groups is unclear. Adding some case studies may help validate the design of the group discovery.
4. In Table 2, the improvement in fairness and task performance is limited and inconsistent. On MedGemma-4B, for fairness-unaware baselines, there is little or no fairness improvement compared with vanilla GRPO in EOD and FPR diff. For fairness-aware baselines, there is little or no task improvement compared with GRPO+DRO in Acc and F1. Some discussion of such observations may be added.

**Questions:**

1. See some questions in Weaknesses.
2. This paper focuses on fairness in medical AI systems, but it gives limited attention to how FairGRPO’s reasoning outputs could integrate with real-world clinical workflows. Including a discussion of interpretability would make the contribution more actionable for medical AI practitioners.
3. I did not find the released FairMedGemma-4B. Maybe I'm missing something?
4. Typo in line 370 "faieness" -> "fairness"

---

> ### Author Response · Authors · 2025-11-21
> **Response to reviewer Tq3Z**
>
> > Tq3Z.C1: The theoretical grounding for the proposed fairness optimization is limited. It is unclear whether the scaling guarantees convergence or prevents overcompensation.
>
> We appreciate the reviewer's question about theoretical guarantees. We clarify that FairGRPO inherits the convergence properties of GRPO while adding demographic-aware scaling. The key insight is that our scaling mechanism preserves the essential structure needed for convergence.
>
> FairGRPO maintains GRPO's advantage normalization structure with $\hat{A}\_{(q,i,t)}^{\text{FairGRPO}} = \frac{s\_{(q,i,t)}^{\text{scaled}}}{\sigma\_{\text{batch}}}$, where the batch-wise renormalization ensures zero mean and unit variance, which is a sufficient condition for GRPO's convergence properties [1]. The demographic scaling factors $T\_{(g,t)}$ and $T\_{(\gamma,g,t)}$ are bounded by design through the square root of sample counts and mean rewards, preventing unbounded scaling that could destabilize training. This bounded scaling ensures that the gradient magnitudes remain within reasonable ranges throughout optimization, preserving the stability properties of the original GRPO algorithm.
>
> In addition, we designed our inverse temperature scaling to prevent overcompensation through its mathematical structure. For a group $\gamma$ with representation $N\_{(\gamma,g,t)}$ and performance $\bar{r}\_{(\gamma,g,t)}$, the scaling factor $T\_{(\gamma,g,t)} = \sqrt{N\_{(\gamma,g,t)} \cdot \bar{r}\_{(\gamma,g,t)}}$ creates a self-regulating mechanism. As minority groups improve (increasing $\bar{r}\_{(\gamma,g,t)}$), their scaling factor increases, reducing the amplification. This creates a natural equilibrium where groups converge toward similar performance levels without overcompensation.
>
> Lastly, while formal convergence proofs for policy gradient methods with adaptive weighting remain an open research question, our extensive experiments across 7 datasets provide strong empirical evidence. Figure 2(a-b) shows that FairGRPO maintains similar convergence trajectories to GRPO for overall metrics while simultaneously improving fairness metrics. The Pareto frontier analysis in Figure 2(e) further demonstrates that FairGRPO finds solutions that dominate GRPO in the performance-fairness space, suggesting the method converges to better optima rather than merely different ones. We acknowledge that providing formal convergence guarantees for adaptive importance weighting in policy optimization is an important direction for future theoretical work, similar to ongoing efforts in the broader fair RL literature.
>
> [1] Zhihong Shao, Peiyi Wang, Qihao Zhu, Runxin Xu, Junxiao Song, Xiao Bi, Haowei Zhang, Mingchuan Zhang, YK Li, Y Wu, et al. "Deepseekmath: Pushing the limits of mathematical reasoning in open language models." https://arxiv.org/abs/2402.03300

---

> ### Author Response · Authors · 2025-11-21
> **Response to reviewer Tq3Z - Continued 1**
>
> > Tq3Z.C2: The paper employs K-means clustering on reward vectors to infer latent demographic groups, but does not analyze the robustness of this clustering step. The number of clusters, initialization, and metric choice could influence group assignments, potentially introducing instability or even amplifying unintended biases in underrepresented groups.
>
> Our hierarchical scaling scheme is specifically designed to address this challenge through several mechanisms. First, as demonstrated in Figure 3, FairGRPO consistently outperforms GRPO on validation F1 scores, particularly for minority groups, indicating robust generalization rather than overfitting to training data. Second, our approach does not simply up-weight minority samples uniformly; instead, it employs a hierarchical structure that considers both dataset-level and demographic-level imbalances, which helps prevent amplification of noise in small populations. In particular, our approach balances the up-weighting through square root scaling of sample sizes in our advantage calculation:
> $$
> T\_{(g,t)} = \sqrt{N\_{(g,t)}} \cdot \bar{r}\_{(g,t)},T\_{(\gamma,g,t)} = \sqrt{N\_{(\gamma,g,t)}} \cdot \bar{r}\_{(\gamma,g,t)}
> $$
> This prevents excessive amplification that could lead to noise overfitting, as the scaling factor grows sublinearly with group size. Lastly, we implement standard gradient clipping as employed in the vanilla GRPO algorithm to stabilize training. Our empirical results across multiple datasets and demographic attributes demonstrate that the scaling mechanism effectively balances improving minority group performance without compromising overall model generalization.
>
> In addition, we employ the elbow method in K-means clustering, which allows the model to decide the optimal number of clusters dynamically throughout the training process. Empirically, we see that the number of discovered clusters for each dataset is always less than 10, even if we set the maximum number of clusters to be unlimited. As a result, assigning a higher maximum cluster count has a negligible effect on the final performance. Having a cluster cap that is too low (i.e., <3), however, does yield degradation in performance of around 5%.
>
> > Tq3Z.C3: The design of group discovery lacks interpretability. The relation between the learned task-specific difficulty patterns and demographic groups is unclear. Adding some case studies may help validate the design of the group discovery.
>
> As a quantitative analysis, when the demographic information is not passed to the model, the discovered groups have an average Adjusted Rand Index (ARI) of 0.109 and a Normalized Mutual Information (NMI) of 0.355, indicating positive correlation between the discovered groups and the ground truth demographic groups.
>
> On the scaling side, we see the model with discovered groups upscale samples from minority groups:
>
> | Demo Group | M,A1 | F,A2 | A2,F | M,A2 | M,A4 |
> |------|------|------|------|------|------|
> | Avg Scale | 0.031±0.037 | 0.078±0.160 | 0.083±0.124 | 0.089±0.170 | 0.110±0.189 |
> | A2,M | F,A4 | F,A3 | F,A1 | M,A3 | A2 |
> | 0.117±0.315 | 0.147±0.219 | 0.153±0.217 | 0.172±0.207 | 0.174±0.239 | 0.641±0.833 |
> | A3 | A3,M | A4 | A1 | A3,F | A1,F |
> | 0.863±1.184 | 1.046±1.110 | 1.414±2.071 | 1.515±1.533 | 2.955±8.435 | 3.024±0.000 |
>
> In the above table, the demographic group is defined as in the paper. For age, we
> create four groups using 25-year bins: a1 for ages 18-25, a2 for ages 26-50, a3 for ages 51-75, and a4 for ages 76 and above. M indicates males, and F denotes females.
>
> We see that even without ground truth demographic groups, the model tends to upscale minority samples, particularly female samples (A1, F scale of 3.024, A3, F scale of 2.955) and people of minority age groups (A1 scale = 1.515, A4 scale = 1.414). This validates our hypothesis that minority samples tend to be harder samples for the model throughout the training process, and upscaling these samples has a positive effect on the fairness of the model.

---

> ### Author Response · Authors · 2025-11-21
> **Response to reviewer Tq3Z - Continued 2**
>
> > Tq3Z.C4: In Table 2, the improvement in fairness and task performance is limited and inconsistent. On MedGemma-4B, for fairness-unaware baselines, there is little or no fairness improvement compared with vanilla GRPO in EOD and FPR diff. For fairness-aware baselines, there is little or no task improvement compared with GRPO+DRO in Acc and F1. Some discussion of such observations may be added.
>
> For the MedGemma model, we would like to point out that our method performs the best among all methods in all performance metrics, 6 out of 7 fairness metrics and all combined metrics. FairGRPO_ND is an ablation of our method where the model tries to improve its performance via clustering on downstream performances, under the assumption that samples from the same demographic group exhibit similar downstream performance. We validated this assumption and evaluated the alignment of the learned cluster and the ground truth cluster in our response to C1 above. In addition, clustering via performance patterns also allows the model to focus more on harder samples, bringing additional performance benefits. On the other hand, FairGRPO evaluates how the model performs when the ground truth demographic information is available. As evidenced in Table 1, compared with fairness-aware baselines, our method is able to leverage the demographic information most effectively, thanks to the design of dynamic weighting. Both methods, as we tested, achieve better Pareto frontier, as the most widely accepted measure for tradeoff between performance and fairness. We will include a more in-depth discussion regarding Pareto frontier within the final camera-ready file:
>
> "In general, it is widely acknowledged within ML fairness literature that outside of special cases,
> it is difficult to simultaneously satisfy multiple definitions of fairness [1,2], thus motivating the need to decide on the best model beyond simple accuracy [2]. FairGRPO is in alignment with this idea as it provides the best results across both performance and fairness as well as the performance-fairness Pareto frontier.  A Pareto frontier can be understood as the set of optimal solutions that strike a balance among different objectives such that there is no better solution beyond the frontier. Moreover, given the constraints, it is noteworthy that our proposed method, FairGRPO, is able to improve fairness across most fairness measures without sacrificing performance on the other metrics.”
>
> [1] Hsu, Brian, et al. "Pushing the limits of fairness impossibility: Who's the fairest of them all?." Advances in Neural Information Processing Systems 35 (2022): 32749-32761
>
> [2] Black, Emily, Manish Raghavan, and Solon Barocas. "Model multiplicity: Opportunities, concerns, and solutions." Proceedings of the 2022 ACM conference on fairness, accountability, and transparency. 2022.

---

> ### Author Response · Authors · 2025-11-21
> **Response to reviewer Tq3Z - Continued 3**
>
> > Tq3Z.C6: I did not find the released FairMedGemma-4B. Maybe I'm missing something?
>
> We have released our code on the anonymous repo https://anonymous.4open.science/r/fairness_submission-D923/, which can be used to reproduce FairMedGemma-4B. The anonymous repo, however, has a file size constraint of 50 MB. Therefore, currently, it is not possible to release the model weights without violating anonymity. We will release the model publicly when the paper is published, and we thank reviewer Tq3Z for understanding regarding this constraint.
>
> > Tq3Z.C7: Typo in line 370 "faieness" -> "fairness"
>
> Thanks for catching this. We have fixed this typo in our new manuscript.

---

### Official Review · Reviewer_cwor · 2025-11-01

**Soundness:** 3
**Presentation:** 3
**Contribution:** 3
**Rating:** 6
**Confidence:** 2

**Summary:**

The paper introduces Fairness-aware Group Relative Policy Optimization (FairGRPO), a reinforcement learning approach designed to train multimodal reasoning foundation models with reduced performance disparities across demographic groups. The method aims to mitigate biases and promote equitable clinical decision-making.

FairGRPO extends GRPO by introducing an adaptive importance-weighting mechanism that normalizes rewards using inverse temperature scaling, where the scaling factor depends on each group’s representation size and performance. Groups can be either explicitly defined (via demographic labels) or implicitly discovered. In the latter case, the authors apply K-means clustering on reward-based representations, selecting the number of clusters using the elbow method. The training objective follows GRPO’s policy gradient formulation with clipped importance sampling.

The authors also release FairMedGemma-4B, a fairness-aware clinical vision–language model trained with FairGRPO. It achieves state-of-the-art diagnostic performance while significantly reducing disparities across demographic groups, as demonstrated on seven clinical datasets spanning five imaging modalities.

**Strengths:**

•	The paper addresses fairness in reinforcement learning for multimodal foundation models, a highly relevant problem in clinical reasoning where demographic disparities can have critical implications.

•	The presentation is clear and easy to follow, with well-explained formulations and experimental design.

•	FairGRPO extends Group Relative Policy Optimization with an adaptive importance-weighting mechanism that normalizes rewards via inverse-temperature scaling based on group size and performance. The approach is conceptually sound and well motivated by recent fairness literature.

•	The authors also consider cases where explicit demographic labels are unavailable, proposing a clustering strategy on reward-based representations to infer latent groups.

•	Experiments on seven clinical datasets spanning five imaging modalities show consistent fairness gains and competitive or superior accuracy. The release of FairMedGemma-4B, a fairness-aware clinical vision–language model, further strengthens the paper’s practical contribution.

**Weaknesses:**

1. The clustering-based grouping used when demographic labels are unavailable is intuitive but not further analyzed. It remains unclear what the clusters capture in practice, or under which conditions they would align with meaningful demographic or clinical subgroups.

2. The proposed FairGRPO objective is reasonable and empirically effective, but a theoretical analysis or discussion of the training objective and convergence behavior would provide a deeper understanding of its effect, rather than relying solely on intuition and empirical evidence.

3. Potential overfitting or generalization issues when up-weighting minority groups are not discussed. It would be helpful to analyze whether the scaling mechanism might amplify noise in small or underrepresented populations.

4. Table 2 does not include standard deviations. Also, a breakdown of performance for each dataset (similar to Table 2 but disaggregated) would improve transparency.

5. The information in Tables 4–17, which report dataset- and group-level metrics, is difficult to parse in its current format, even though these results seem to be relevant to understand the performance at a dataset/group level for each method. Could these be summarized?

**Questions:**

Please, see weaknesses.

---

> ### Author Response · Authors · 2025-11-21
> **Response to reviewer cwor**
>
> > cwor.C1: The clustering-based grouping used when demographic labels are unavailable is intuitive but not further analyzed. It remains unclear what the clusters capture in practice, or under which conditions they would align with meaningful demographic or clinical subgroups.
>
> We thank reviewers for their interest in an analysis of how the cluster is aligned with the ground truth cluster. As a quantitative analysis, when the demographic information is not passed to the model, the discovered groups have an average Adjusted Rand Index (ARI) of 0.109 and a Normalized Mutual Information (NMI) of 0.355, indicating positive correlation between the discovered groups and the ground truth demographic groups.
>
> On the scaling side, we see the model with discovered groups upscale samples from minority groups:
>
> | Demo Group | M,A1 | F,A2 | A2,F | M,A2 | M,A4 |
> |------|------|------|------|------|------|
> | Avg Scale | 0.031±0.037 | 0.078±0.160 | 0.083±0.124 | 0.089±0.170 | 0.110±0.189 |
> | A2,M | F,A4 | F,A3 | F,A1 | M,A3 | A2 |
> | 0.117±0.315 | 0.147±0.219 | 0.153±0.217 | 0.172±0.207 | 0.174±0.239 | 0.641±0.833 |
> | A3 | A3,M | A4 | A1 | A3,F | A1,F |
> | 0.863±1.184 | 1.046±1.110 | 1.414±2.071 | 1.515±1.533 | 2.955±8.435 | 3.024±0.000 |
>
> In the above table, the demographic group is defined as in the paper. For age, we
> create four groups using 25-year bins: a1 for ages 18-25, a2 for ages 26-50, a3 for ages 51-75, and a4 for ages 76 and above. M indicates males, and F denotes females.
>
> We see that even without ground truth demographic groups, the model tends to upscale minority samples, particularly female samples (A1, F scale of 3.024, A3, F scale of 2.955) and people of minority age groups (A1 scale = 1.515, A4 scale = 1.414). This validates our hypothesis that minority samples tend to be harder samples for the model throughout the training process, and upscaling these samples has a positive effect on the fairness of the model.
>
> > cwor.C2: The proposed FairGRPO objective is reasonable and empirically effective, but a theoretical analysis or discussion of the training objective and convergence behavior would provide a deeper understanding of its effect, rather than relying solely on intuition and empirical evidence.
>
> We will include the following discussion within the final camera-ready file:
>
> “The FairGRPO objective constitutes a modification of the original GRPO objective such that the model is trained to update the policy model in the direction of high advantage scaled across demographic groups and task difficulty. Intuitively, this means that the original GRPO advantages (e.g. simplification of the training process without the need for a value model) is preserved. Therefore, the convergence behaviour of FairGRPO will be similar to that of the GRPO objective, with the key difference that it will be scaled across demographic groups and task difficulty, thus converging towards a solution that improves both model performance and fairness, which is a trend that we have seen in our results (as evidenced by the results in Table 2 and the Pareto frontier in Figure 2). Empirically, we see evidence of this in Figure 2(a) and (b), where the convergence behaviour of FairGRPO is similar to that of GRPO.
>
> > cwor.C3: Potential overfitting or generalization issues when up-weighting minority groups are not discussed. It would be helpful to analyze whether the scaling mechanism might amplify noise in small or underrepresented populations.
>
> Our hierarchical scaling scheme is specifically designed to address this challenge through several mechanisms. First, as demonstrated in Figure 3, FairGRPO consistently outperforms GRPO on validation F1 scores, particularly for minority groups, indicating robust generalization rather than overfitting to training data. Second, our approach does not simply up-weight minority samples uniformly; instead, it employs a hierarchical structure that considers both dataset-level and demographic-level imbalances, which helps prevent amplification of noise in small populations. In particular, our approach balances the up-weighting through square root scaling of sample sizes in our advantage calculation:
> $$
> T\_{(g,t)} = \sqrt{N\_{(g,t)}} \cdot \bar{r}\_{(g,t)},T\_{(\gamma,g,t)} = \sqrt{N\_{(\gamma,g,t)}} \cdot \bar{r}\_{(\gamma,g,t)}
> $$
> This prevents excessive amplification that could lead to noise overfitting, as the scaling factor grows sublinearly with group size. Lastly, we implement standard gradient clipping as employed in the vanilla GRPO algorithm to stabilize training. Our empirical results across multiple datasets and demographic attributes demonstrate that the scaling mechanism effectively balances improving minority group performance without compromising overall model generalization.

---

> ### Author Response · Authors · 2025-11-21
> **Response to reviewer cwor - Continued 1**
>
> > cwor.C4: Table 2 does not include standard deviations. Also, a breakdown of performance for each dataset (similar to Table 2 but disaggregated) would improve transparency. The information in Tables 4–17, which report dataset- and group-level metrics, is difficult to parse in its current format, even though these results seem to be relevant to understand the performance at a dataset/group level for each method. Could these be summarized?
>
> We agree that multiple runs are useful to evaluate the significance of results. Initial experiments were conducted once per method, mainly due to computational constraints, as the model takes multiple days to train on 4 NVIDIA H200 GPUs. We further ran each experiment 2 times and included the results in the table below:
>
> | Training Method | **Fairness Metrics** | | | | | | | **Perf. Metrics** | | **Combined** | |
> |---|---|---|---|---|---|---|---|---|---|---|---|
> | | PP ↓ | EOD ↓ | FPR_Diff ↓ | σ_F1 ↓ | ΔF1 ↓ | σ_Acc ↓ | ΔAcc ↓ | Acc ↑ | F1 ↑ | Acc_ES ↑ | F1_ES ↑ |
> | **Qwen-2.5-VL-7B** ||||||||||||
> | Re++ | 16.66 ± 2.11 | 6.66 ± 1.59 | 6.37 ± 0.20 | .0322 ± .0000 | .0647 ± .0004 | 5.06 ± 0.49 | 10.33 ± 1.01 | 75.31 ± 1.82 | .2599 ± .0065 | 71.69 ± 1.39 | .2518 ± .0063 |
> | RLOO | 22.34 ± 0.86 | 6.67 ± 0.13 | 5.68 ± 0.80 | .0330 ± .0006 | .0693 ± .0017 | 4.86 ± 0.33 | 10.00 ± 0.79 | 78.22 ± 0.06 | .2523 ± .0013 | 74.59 ± 0.18 | .2443 ± .0014 |
> | GRPO | 17.90 ± 9.21 | 7.93 ± 1.64 | **4.85 ± 0.34** | .0387 ± .0107 | .0821 ± .0215 | 4.85 ± 0.24 | 9.92 ± 0.69 | 78.40 ± 0.69 | .2601 ± .0131 | 76.21 ± 0.91 | .2425 ± .0017 |
> | GRPO+RS | 19.62 ± 7.22 | 6.85 ± 0.80 | 6.44 ± 1.39 | .0319 ± .0009 | .0628 ± .0037 | 5.50 ± 0.17 | 11.26 ± 0.34 | 75.61 ± 2.96 | .2580 ± .0021 | 71.67 ± 2.92 | .2500 ± .0018 |
> | FairGRPO | **15.42 ± 1.95** | **5.62 ± 0.10** | 5.00 ± 0.87 | **.0254 ± .0035** | **.0522 ± .0099** | **4.42 ± 0.01** | **8.95 ± 0.03** | **78.52 ± 0.31** | **.2657 ± .0036** | **77.14 ± 0.29** | **.2602 ± .0020** |
> | **MedGemma-4B** ||||||||||||
> | Re++ | 20.30 ± 0.97 | 7.78 ± 1.37 | 5.69 ± 0.10 | .0469 ± .0069 | .0898 ± .0191 | 4.44 ± 0.17 | 8.99 ± 0.25 | 78.76 ± 0.22 | .3105 ± .0179 | 75.41 ± 0.09 | .2966 ± .0191 |
> | RLOO | 20.45 ± 4.57 | 10.35 ± 0.03 | 5.51 ± 0.01 | .0592 ± .0011 | .1173 ± .0004 | 4.29 ± 0.07 | 8.79 ± 0.06 | 79.76 ± 0.16 | .3237 ± .0019 | 76.48 ± 0.20 | .3056 ± .0021 |
> | GRPO | 20.89 ± 2.16 | **6.30 ± 0.25** | 5.26 ± 0.62 | .0387 ± .0045 | .0753 ± .0059 | 4.19 ± 0.03 | 8.57 ± 0.03 | 79.38 ± 0.15 | .3134 ± .0118 | 76.19 ± 0.12 | .3017 ± .0101 |
> | GRPO+RS | 24.55 ± 1.12 | 6.97 ± 0.44 | **4.78 ± 1.84** | .0422 ± .0017 | .0834 ± .0003 | 4.20 ± 0.21 | 8.77 ± 0.54 | 79.02 ± 0.15 | .2825 ± .0052 | 75.84 ± 0.30 | .2711 ± .0046 |
> | GRPO+DRO | 18.20 ± 3.06 | 7.52 ± 0.22 | 5.68 ± 0.98 | .0456 ± .0013 | .0895 ± .0034 | 4.55 ± 0.26 | 9.39 ± 0.61 | 80.17 ± 0.31 | .3146 ± .0177 | 76.69 ± 0.48 | .3009 ± .0173 |
> | FairGRPO_ND | 24.87 ± 0.40 | 9.09 ± 3.49 | 6.35 ± 0.93 | .0484 ± .0088 | .0919 ± .0210 | 4.18 ± 0.80 | **8.36 ± 1.62** | 78.82 ± 0.58 | **.3484 ± .0041** | 75.67 ± 1.14 | **.3323 ± .0011** |
> | FairGRPO | **12.95 ± 1.82** | 6.84 ± 0.24 | 5.53 ± 0.29 | **.0379 ± .0005** | **.0724 ± .0004** | **4.11 ± 0.04** | 8.53 ± 0.11 | **80.40 ± 0.03** | .3275 ± .0007 | **77.23 ± 0.01** | .3155 ± .0006 |
>
> In general, we see our method yields consistent and significant improvements over baselines, with σ_F1 (p=0.000), ΔF1 (p=0.048), σ_Acc (p=0.001), ΔAcc (p=0.005), Acc_ES (p=0.045), and F1_ES (p=0.010) for Qwen-2.5-VL-7B and PP (p=0.003), EOD (p=0.001), σ_Acc (p=0.002), F1 (p=0.000), Acc_ES (p=0.028), and F1_ES (p=0.000) for MedGemma-4B all being statistically significant.
>
> Furthermore, we calculated the results with the same three runs. To avoid flooding the rebuttal page, the per-dataset results are included as Appendix Table 19-24 in the new manuscript. We hope these results provide a clearer comparison of our methods against other baselines.

---

### Meta-Review · Area_Chair_rjp2 · 2025-12-16

**Summary:**

This paper proposes FairGRPO, a fairness-aware reinforcement learning method for training multimodal clinical vision-language models that addresses demographic disparities in medical AI. The method employs adaptive importance weighting based on group representation, task difficulty, and data source, and includes unsupervised clustering using K-means to discover latent demographic groups when labels are unavailable.

The primary strengths include addressing an important and underexplored problem—fairness in RL for clinical models where demographic disparities have critical real-world implications. The method is conceptually sound with adaptive importance weighting that normalizes rewards via inverse-temperature scaling, and experiments span 7 clinical datasets across 5 imaging modalities demonstrating consistent fairness improvements. The release of FairMedGemma-4B as a fairness-aware clinical VLLM strengthens practical contribution, and training dynamics analysis shows FairGRPO progressively improves fairness while baseline RL methods degrade it.

However, reviewers raised concerns about limited technical novelty in the proposed method, as the use of importance weighting to mitigate demographic disparity is very classic [1] and many others.

[1]. Fairness without Demographics through Adversarially Reweighted Learning, NeurIPS 2020

Furthermore, when group labels are not available, I agree with the reviewers that the clustering approach for discovering demographic groups is not deeply analyzed regarding robustness, interpretability, or conditions under which clusters align with meaningful subgroups. Some experimental results show modest and inconsistent improvements, particularly on MedGemma-4B where fairness-unaware baselines show little improvement in some metrics (EOD, FPR diff) compared to vanilla GRPO. Reviewers also noted insufficient novelty concerns—the modification of advantages is relatively straightforward. Finally, the connection between group-level fairness improvements and individual-level fairness metrics needs clearer articulation for clinical applicability.

**Reviewer Concerns:**

Addressed: TS5D
Outstanding: the remaining

**Reviewer Scores:**

hard to calibrate.

---

### Decision · Program_Chairs · 2026-01-26

Reject